1 Joint characterization of heterogeneous conductivity fields and pumping well attributes through iterative ensemble smoother with a reduced-order modeling 2 strategy for solute transport 3 4 Chuan-An Xia<sup>1</sup>, Jiayun Li<sup>2\*</sup>, Bill X. Hu<sup>3</sup>, Alberto Guadagnini<sup>4,5</sup>, Monica Riva<sup>4\*</sup> 5 6 <sup>1</sup>Zijin School of Geology and Mining, Fuzhou University, Fuzhou, China 7 <sup>2</sup>Fujian Provincial Key Lab of Coastal Basin Environment, Fujian Polytechnic Normal 8 University, Fuqing, China 9 <sup>3</sup>School of Water Conservancy & Environment, University of Jinan, Jinan, China 10 <sup>4</sup>Dipartimento di Ingegneria Civile e Ambientale, Politecnico di Milano, Milano, Italy 11 12 <sup>5</sup>Sonny Astani Department of Civil and Environmental Engineering, Viterbi School of Engineering, Los Angeles, California 90089-2531, USA 13 14 15 16 17 18 19 Corresponding author: Jiayun Li; Monica Riva Email: lijy@fpnu.edu.cn; monica.riva@polimi.it 20 21

22 Abstract

23 We develop and test an efficient and accurate theoretical and computational framework to jointly estimate spatially variable hydraulic conductivity and identify 24 unknown pumping well locations and rates in a two-dimensional confined aquifer. 25 26 The approach (denoted as iES\_ROM) integrates an iterative Ensemble Smoother (iES) with a Reduced-Order Model (ROM) for solute transport taking place across an 27 28 otherwise steady-state groundwater flow field. This offers a computationally efficient alternative to the Full System Model (iES\_FSM) upon addressing the high 29 30 computational demands of ensemble-based data assimilation methods, which typically require large ensemble sizes to characterize uncertainties in (randomly) heterogeneous 31 aquifers. Our iES\_ROM is constructed through proper orthogonal decomposition. It is 32 33 then evaluated across a collection of 28 test cases exploring variations in model 34 dimension, ensemble size, measurement noise, monitoring network, and statistical properties of the (underlying randomly heterogeneous) conductivity field. Our results 35 support the ability of iES\_ROM to accurately estimate conductivity and identify 36 37 pumping well attributes under diverse configurations, attaining a quality of performance similar to iES\_FSM. When using moderate ROM dimensions (n = 25-30) 38 and ensemble size (i.e., 500-1000), the accuracy of iES\_ROM does not vary 39 significantly while computational time is reduced by nearly an order of magnitude. 40 41 Our approach thus provides a reliable and cost-effective tool for inverse modeling in groundwater systems with uncertain parameters. 42 Keywords: reduced-order model; proper orthogonal decomposition; iterative 43

https://doi.org/10.5194/egusphere-2025-5320 Preprint. Discussion started: 7 November 2025 © Author(s) 2025. CC BY 4.0 License.

ensemble smoother; pumping well identification; groundwater

#### 1. Introduction

45

Assessment of groundwater flow and transport scenarios is typically plagued by uncertainties associated with model structure and parametrization. A major source of 47 uncertainty often examined concerns the poorly constrained assessment of pollution 48 49 sources. Our ability to identify spatial locations of these sources exerts significant influence on the design of contaminant monitoring, management, and remediation 50 51 strategies. Contaminant release to an aquifer is characterized through the spatial 52 location of sources, the temporal variability of release fluxes, and solute 53 concentrations involved (Chen et al., 2018; Xu et al., 2018; Mo et al., 2019). Uncertainties linked to groundwater abstraction scheduling also play a critical role, as 54 operational details of pumping wells are not always fully documented. For example, 55 this might correspond to a scenario where such information is not disclosed to ensure 56 57 privacy protection or uncertainties are induced by geocoding practices and/or measurement devices. Further to this, in some regions groundwater may be accessed 58 through wells that are not officially registered or fully documented by industrial 59 60 operators and/or local residents. Despite the relevance of these issues, only limited research has been devoted to the identification and quantification of pumping rates 61 and spatial locations of such hidden wells. 62 In this broad context, we recall that a considerable body of research has focused 63 64 on estimating key parameters (such as hydraulic conductivity) in groundwater flow and transport models through ensemble-based Data Assimilation (DA) techniques 65 (Chen and Zhang, 2006; Tong et al., 2013; Chen and Oliver, 2013; Zhang et al., 2018; 66

Xia et al., 2018, 2024). These approaches aim at enhancing the accuracy of simulated 68 system states (e.g., hydraulic heads and solute concentrations). While their capability to jointly estimate parameters and update system states has been broadly explored, 69 their high computational cost still constitutes a persistent limitation to their practical 70 71 routine use. This challenge primarily stems from the requirement for a large number of realizations to ensure statistical convergence of Monte Carlo (MC) simulations 72 73 (e.g., Ballio and Guadagnini, 2004) in the forecast step of the DA process, and to 74 achieve reliable parameter estimates in the analysis step. The computational burden 75 becomes particularly significant when the selected model describing the system behavior (hereafter termed as Full System Model (FSM)) must be repeatedly executed 76 for systems characterized by strong nonlinearities or requiring high (space-time) 77 78 resolution of state variables and parameters. 79 To alleviate these computational constraints, recent studies explore the benefit of relying on surrogate (or reduced-order) models that approximate the behavior of the 80 full system while maintaining sufficient accuracy for inverse modeling workflows and 81 82 Uncertainty Quantification (UQ). In this framework, efforts to mitigate computational limitations of 83 (ensemble-based) DA methods primarily focus on the adoption of localization 84 techniques (e.g., Xia et al., 2018, 2024; Luo and Bahkta, 2020) or surrogate modeling 85 86 strategies (e.g., Zhang et al., 2018; Mo et al., 2019 and references therein). Main advantages associated with location approaches are related to the observation that they (i) substantially reduce computational costs upon requiring only a limited 88

accuracy of the assimilated results, and (ii) retain a physically-based and 90 mathematically-tractable formulation. As a notable drawback of these approaches, we 91 note that the value of the information associated with diverse measurements may be 92 93 partially suppressed due to the use of distance- or correlation-based localization, which might constrain the strength of the spatial influence of observations. As a 94 95 consequence, the ensuing (empirical/sample) probability density functions (PDFs) of 96 model parameters and system states often display reduced accuracy and fail to fully 97 capture the underlying uncertainty structure. To mitigate these limitations, an alternative line of research explores the use of surrogate models (SMs), which aim at 98 emulating the response of the Full System Model with significantly reduced 99 100 computational cost while preserving the salient physics of the system. 101 Surrogate models are rapidly emerging as a promising complement to FSMs for reducing computational burdens associated with the forecast steps of ensemble-based 102 DA procedures. Among the various SM strategies, data-driven approaches based on 103 104 machine learning (e.g., Ju et al., 2018) and deep learning (e.g., Mo et al., 2019) can be employed for emulating groundwater flow and transport processes taking place in 105 heterogeneous media. For example, Ju et al. (2018) rely on Gaussian Process 106 regression to describe relationships between the coefficients of a Karhunen-Loève 107 108 (KL) expansion (employed to characterize a spatially heterogeneous hydraulic 109 conductivity field) and (point-wise) simulated observations. This approach is shown to achieve approximately an order of magnitude reduction in computational time as 110

number of Monte Carlo realizations of the FSM, while maintaining acceptable

compared with the standard iterative Ensemble Smoother (iES). Otherwise, this gain 111 112 in efficiency is associated with a reduced accuracy in simulated hydraulic heads, which in turn compromises the reliability of the estimated conductivity field. Mo et al. 113 (2019) employ deep autoregressive neural networks as an FSM surrogate to 114 115 reconstruct conductivity fields and identify contaminant source characteristics. However, their approach still requires a significant computational effort, as it heavily 116 117 relies on a high number (about 1,500 in their exemplary setting) of MC realizations of 118 the FSM for network training. While these studies show a clear potential of 119 data-driven surrogates for accelerating DA workflows, they also highlight the need for a fundamental trade-off between computational efficiency and model accuracy, thus 120 underscoring the potential value of alternative surrogate modeling strategies. 121 122 In contrast to data-driven models, that typically operate as black-box 123 representations, projection-based Reduced-Order Models (ROMs) are physics-based (e.g., Razavi et al., 2012; Asher et al., 2015; Chen et al., 2017; Xia et al., 2020, 2025). 124 ROMs are typically constructed upon projecting the governing equations and 125 126 boundary conditions of the onto a lower-dimensional subspace spanned by a set of basis functions. The latter are commonly derived through, e.g., Proper Orthogonal 127 Decomposition (POD) of multiple FSM solutions, referred to as snapshots. This 128 procedure effectively reduces the dimensionality of the system state space. The 129 130 random field representing the system state can then be expressed as a linear combination of the dominant eigenfunctions obtained from the Fredholm integral 131 equation associated with the covariance matrix of the snapshots. Leading 132

eigenfunctions are then identified as the basis functions defining the reduced subspace. 133 134 Substantial computational savings are then achieved upon resting on the solution of the ensuing low-dimensional linear system. When implemented in the context of 135 numerical MC frameworks, the collection of ROM-generated solutions constitutes 136 137 what is commonly referred to as a Reduced-Order Monte Carlo (ROMC) simulation framework. 138 139 Reduced-order modeling has received growing attention in the context of 140 groundwater flow (Pasetto et al., 2011, 2013, 2014; Li et al., 2013a; Boyce et al., 2015; 141 Stanko et al., 2016; Xia et al., 2020, 2025) and solute transport (Luo et al., 2012; Li et al., 2013b; Rizzo et al., 2018) scenarios. Its potential is evidenced across a wide range 142 of hydrogeological configurations, including confined (e.g., Pasetto et al., 2011) and 143 unconfined (e.g., Stanko et al., 2016) aquifer systems, homogeneous (e.g., Li et al., 144 145 2013a) and heterogeneous (e.g., Pasetto et al., 2013) media, as well as scenarios with (e.g., Xia et al., 2020) or without (e.g., Pasetto et al., 2014) pumping wells operating 146 therein. Several studies further advance development of ROMC strategies for UQ in 147 148 groundwater flow modeling. Pasetto et al. (2014) show that the accuracy of UQ results relying on ROMC in the presence of steady-state groundwater flow strongly 149 depends on the quality and the number of snapshots, the latter directly influencing 150 representativeness of the basis functions. To mitigate this limitation, Xia et al. (2020) 151 152 propose deriving basis functions as the leading eigenvectors of (second-order) approximations of hydraulic head covariances. The latter are obtained upon solving 153 the associated moment equations for steady-state groundwater flow (Zhang and Lu, 154

2002; Xia et al., 2019). Even as reduction of the dimensionality of the head space provides substantial computational savings, projection of the basis functions onto the ensuing (typically large) system matrix remains computationally intensive, thereby still constituting a limiting factor to efficiency gains. Xia et al. (2025) address this challenge by extending their approach to perform dimensionality reduction for both (spatially variable) transmissivity and hydraulic head fields in a steady-state groundwater flow setting and achieving additional computational savings while maintaining high accuracy. Despite these advancements, most existing ROM and ROMC approaches are still fraught with difficulties in efficiently capturing strongly nonlinear system dynamics and adapting to evolving state conditions, underscoring the need for more flexible and computationally efficient reduced-order frameworks. With reference to solute transport, ROMs have been developed for both homogeneous (Luo et al., 2012) and heterogeneous (Li et al., 2013b; Rizzo et al., 2018) aquifer systems. Li et al. (2013a) further consider construction of ROMs to tackle density-dependent groundwater flow taking place across homogeneous and heterogeneous domains. Otherwise, studies explicitly focusing on the development of ROMC approaches for UQ of solute transport remain limited. Although conceptual insights can be drawn from ROMC studies addressing groundwater flow (e.g., Pasetto et al., 2014; Xia et al., 2020, 2025), influence of key factors (such as, e.g., dimensionality of the reduced concentration space and strength of hydraulic conductivity heterogeneity) on accuracy and robustness of ROMC-based UQ still remains poorly characterized.

Building upon these works, the present study introduces a novel framework that integrates the iES with a ROM for solute transport (hereafter referred to as iES ROM). The ensuing framework enables one to efficiently quantify uncertainty and jointly estimate system parameters in groundwater-related modeling scenarios. The proposed method is then applied to simultaneously identify pumping rate and spatial location of (otherwise hidden) wells operating within the system, while providing estimates of the spatially heterogeneous hydraulic conductivity field under conditions of steady-state flow and transient solute transport. In the iES\_ROM framework, the steady-state flow field is evaluated through the FSM, whereas the transient solute transport is represented by a computationally efficient ROM. The required snapshots and associated POD are generated only once. These are subsequently employed throughout the entire DA process, thus avoiding repeated high-fidelity simulations. To ensure transparent benchmarking, the performance of iES ROM is systematically compared with that of a reference approach (termed iES\_FSM) which relies entirely on the FSM associated with synthetic scenarios. Comparative analyses are performed across a variety of synthetic scenarios, encompassing diverse ROM dimensions, ensemble sizes, measurement qualities and quantities, as well as distinct statistical descriptors of the initial conductivity ensemble and snapshot sizes. The study is organized as follows. Section 2 introduces the theoretical background of groundwater flow and solute transport and details the integration of ROMC simulation within the iES framework. Section 3 describes the test cases

- designed to evaluate the proposed approach. Section 4 illustrates and discusses the
- main results, and Section 5 summarizes the key findings.

# 201 2. Theory background and methodology

## 2.1 Groundwater flow and solute transport

We consider two-dimensional steady-state groundwater flow governed by:

$$\nabla \cdot \left[ K(\mathbf{x}) \nabla h(\mathbf{x}) \right] + q_s(\mathbf{x}) = 0$$
 (1)

- where  $\mathbf{x} = [x_1, x_2]$  is a vector of spatial coordinates in domain  $\Omega^2$ ; h is hydraulic
- head; K is (isotropic) hydraulic conductivity; and  $q_s$  is a source/sink term. We
- conceptualize K as a spatially heterogeneous random field, associated with a given
- spatial correlation structure. The source/sink term in Equation (1) corresponds to a
- production well associated with an uncertain pumping rate and location in the domain.
- Propagation of uncertainty related to model parameters and/or forcing terms onto
- hydraulic heads and fluxes is typically assessed through numerical Monte Carlo (MC)
- simulations (see, e.g., Ballio and Guadagnini, 2004; Xia et al., 2020, 2024, and
- references therein).
- We consider (non-reactive) solute transport evolving in  $\Omega^2$  to be described
- through:

$$\nabla \cdot \left[ D\nabla c(\mathbf{x}, t) \right] - \nabla \left( \mathbf{q}(\mathbf{x}) c(\mathbf{x}, t) \right) + \frac{q_s(\mathbf{x})}{\theta} c_s(\mathbf{x}, t) = \frac{\partial c(\mathbf{x}, t)}{\partial t}$$
 (2)

- Here, t denotes time; c is solute concentration; D is the (isotropic) dispersion
- coefficient;  $\theta$  is effective porosity;  $c_s$  is solute concentration corresponding to  $q_s$ ;
- and  $\mathbf{q}(\mathbf{x}) = -(K(\mathbf{x})/\theta)\nabla h(\mathbf{x})$  is an effective velocity associated with solute
- transport.

Numerical methods (e.g., finite differences or finite elements) are commonly 221 222 employed to discretize Equations (1) and (2) that are then solved within a numerical MC context. The probability distribution of state variables of interest (e.g., heads or 223 concentrations) is then evaluated at N nodes of an aptly designed numerical grid. 224 225 Consistent with Section 1, we refer to the model corresponding to the numerical solution of the above equations as the Full System Model (FSM). When the domain is 226 227 characterized by a large spatial extent and/or one is interested in exploring the system 228 behavior across long temporal windows, performing numerical MC simulations 229 relying on FSM is associated with a heavy computational burden. To circumvent this issue, we rely on the development and implementation of a Reduced-Order Model 230 (ROM) strategy for solute transport. We note that in this study we employ ROM 231 232 solely for solute transport because only limited computational costs are associated 233 with the steady-state flow condition we consider, as opposed to simulating transport. Hereafter, we refer to numerical MC analyses grounded on ROM as ROMC. 234 235

# 2.2 Numerical Monte Carlo simulation framework for solute transport

#### 236 2.2.1 Monte Carlo simulation setting for the Full System Model

We rely on a standard finite element method to solve the FSM described in Section 2.1. When considering a total simulation time  $T_s$ , we express the linear system associated with the numerical solution of solute transport through FSM within time interval  $[t, t + \Delta t]$  as:

$$\mathbf{A}^{i}\mathbf{c}^{i} = \mathbf{F}^{i} \tag{3}$$

Here, superscript i refers the to the i<sup>th</sup> MC realization ( $i = 1, ..., N_{MC}, N_{MC}$  being the 242

- total number of MC simulations) of FSM; A is the full-system stiffness matrix (of
- size  $N \times N$ ; **c** is the vector (of size  $N \times 1$ ) of solute concentration values; and **F** is the
- stress vector (of size  $N \times 1$ ) whose entries encompass source/sink terms and initial and
- boundary conditions.

### 2.2.2 Reduced-order Monte Carlo simulation framework

- We construct a reduced-order model for solute transport by approximating the
- solution of solute concentration for the  $i^{th}$  MC realization of FSM. Consistent with the
- work of Xue and Xie (2007) and Pinnau (2008), one can approximate  $\mathbf{c}^i$  as:

$$\mathbf{c}^i \approx \sum_{j=1}^n \alpha_j^i \mathbf{p}_j = \mathbf{P} \mathbf{\alpha}^i$$
 (4)

- Here,  $\mathbf{P} = [\mathbf{p}_1, \mathbf{p}_2, \dots, \mathbf{p}_n]$  is a matrix (of size  $N \times n$ , n being the dimension of the ROM)
- collecting the n nodal basis functions that are here obtained through a Proper
- Orthogonal Decomposition (POD) approach (see below);  $\mathbf{u}^i = \left[\alpha_1^i, \alpha_2^i, \dots, \alpha_n^i\right]^T$  (T
- representing transpose) is a vector (of size  $n \times 1$ ) of Fourier coefficients (Pinnau, 2008).
- Note that Equation (4) is different from a typical Karhunen-Loève expansion of  $c^i$
- (i.e.,  $\mathbf{c}^i \approx \langle \mathbf{c} \rangle + \sum_{j=1}^n \alpha_j^i \mathbf{p}_j = \langle \mathbf{c} \rangle + \mathbf{P} \alpha^i$ , see Equation (11) in Li et al., 2013b). As we
- illustrate in Section 2.3, relying on Equation (4) enables straightforward (i) coding
- and (ii) compatibility with the iterative Ensemble Smoother (iES).
- Substituting Equation (4) into Equation (3) and imposing the residual of the
- model equation associated with the approximated solution to be orthogonal to the
- projection space defined through **P** yields:

$$\mathbf{P}^{T}\mathbf{A}^{i}\mathbf{P}\mathbf{\alpha}^{i} \approx \mathbf{P}^{T}\mathbf{F}^{i} \tag{5}$$

Solving Equation (5) (which is a linear system of size n) yields  $\alpha^i$  for the i<sup>th</sup>

- MC realization of our ROMC strategy. Note that, when  $n \ll N$ , the computational
- effort required by our ROMC is much less than that of the standard MC.
- The basis functions forming the entries of **P** are computed as from the leading
- eigenvectors (corresponding to the highest eigenvalues) of the covariance of totally
- $N_{sn}$  numerical solutions (i.e.,  $\mathbf{c}^1$ ,  $\mathbf{c}^2$ , ..., and  $\mathbf{c}^{N_{sn}}$ ) through FSM. We point out
- that  $N_{sn} = m \times N_t$ , where m is the number of MC realizations which are arbitrarily
- chosen, each yielding  $N_t = T_s / \Delta t$  numerical solutions of Equation (2), with
- uniform length of time step  $\Delta t$ . The leading eigenvectors are computed through the
- Singular Value Decomposition (SVD) approach, i.e.:

$$\mathbf{U}\Lambda\mathbf{U}^{T} = svd\left(\mathbf{E}\mathbf{E}^{T}\right) \tag{6}$$

- where  $\mathbf{E} = 1/\sqrt{N_{sn}} \left[ \mathbf{c}^1, \mathbf{c}^2, \dots, \mathbf{c}^{N_{sn}} \right]$ ;  $\mathbf{U}$  (of size  $N \times N$ ) is the left singular matrix
- whose  $j^{th}$  column is the  $j^{th}$  eigenvector of matrix  $\mathbf{E}\mathbf{E}^{T}$  corresponding to the  $j^{th}$  singular
- value,  $\lambda_j$ ; and  $\Lambda = diag([\lambda_1, \lambda_2, \dots, \lambda_N])$  whose entries are ranked in descending
- order.

# 2.3 Iterative ensemble smoother

- We denote by  $\mathbf{m} = [Y_1, Y_2, ..., Y_N, \ln q_s, x_{1,q_s}, x_{2,q_s}]^T$  the vector (of size P = N+3)
- whose entries correspond to the uncertain model parameters (i.e., the log-conductivity,
- $Y = \ln K$ , field) and flow rate and location of a pumping well. In case the pumping rate
- and location are known, then  $\mathbf{m} = [Y_1, Y_2, \dots, Y_N]^T$  and P = N. We further denote by
- $\mathbf{d} = [d_1, d_2, ..., d_O]^T$  the vector (of size O) of the observations (i.e., measured head and
- concentration values). To estimate m, we implement the iES (Luo and Bhakta, 2020;
- Xia et al., 2024):

approximating solute concentration via Equation (4), we only obtain m numerical

solutions of solute concentration through FSM at the first outer iteration of iES.

Leading eigenvectors are computed upon relying on these solutions and are then

stored. The Fourier coefficients  $\alpha^i$  associated with time interval  $[t, t + \Delta t]$  for each

MC realization starting from the second outer iteration of iES are obtained solely

through solving Equation (5).

When implementing the LM algorithm during optimization, we set both the inner

and outer iteration numbers equal to 10 (see also Luo and Bhakta, 2020). Additionally,

a stopping criterion  $\left(\delta_{k-1} - \delta_k\right) / \delta_{k-1} \times 100\% \le 10^{-6}$  (where

$$\delta_{k} = \frac{1}{N} \sum_{j=1}^{N} \left\{ \left( \mathbf{d}_{j}^{k} - g\left( \mathbf{m}_{j}^{k} \right) \right)^{T} \underline{\underline{\mathbf{C}}}_{d}^{-1} \left( \mathbf{d}_{j}^{k} - g\left( \mathbf{m}_{j}^{k} \right) \right) \right\}, \text{ is set.}$$

# 2.4 Computational cost

We denote by iES\_FSM and iES\_ROM the approaches associated with coupling the iES with FSM and ROM, respectively. The total number of MC realizations is denoted as  $N_{MC}$ . Neglecting the computational cost of the inner iterations and assuming iES comprises  $N_{out}$  outer iterations, the main computational costs of either method can be divided into two components, corresponding to forecast and analysis step (Table 1), respectively. In the forecast step, a number of  $(N_{out}+1)$  MC simulations for groundwater flow and solute transport are required. Otherwise, Equation (7) is evaluated  $N_{out}$  times in the analysis step. The steady-state groundwater flow is solved through the FSM in both iES\_FSM and iES\_ROM, with a computational cost of order  $O(N^3N_{MC})$ . The main computational cost for the  $N_{MC}$  FSM-based MC realizations of solute transport at a single time step in iES\_FSM is

 $O(N^3 N_{MC})$ , while being  $O((sN + N^2) N_{MC})$  (where  $s \approx 7$  or  $\approx 15$  in two and three 328 329 dimensions, respectively) for iES ROM. These computational efforts correspond to the projection of the full-system stiffness matrix onto the reduced-order space of the 330 system state (i.e., solute concentration). Computational costs associated with solving 331 332 Equation (7) coincide for both approaches and are here denoted as  $C_8$ . We further note that, with reference to iES\_ROM, the  $N_{sn}$  solutions of solute concentration obtained 333 through FSM (associated with a computational cost of order  $O(N^3N_{sn})$ ) and the 334 basis functions obtained through SVD (with a computational cost of order  $O(nN_{sn}^2)$ ) 335 336 are calculated only once and stored. When the grid mesh employed is large or the simulation time is long, computational savings through iES\_ROM compared with 337 iES FSM become significant. 338

## 3. Exemplary scenarios

We consider a two-dimensional computational domain of size  $4 \times 2$  to simulate a synthetic sandbox-scale experiment where (non-reactive) solute transport under steady-state flow is considered (see Fig. 1). Here and hereafter, all quantities are given in consistent (length/mass/time) units. Concerning groundwater flow, the left and right sides of the domain are associated with constant head boundary conditions with H = 3 and 2, respectively. The top and bottom sides correspond to boundary conditions of no flow. A pumping well with an unknown pumping rate and location is considered in the setting. A fixed concentration boundary is set at point (0, 1) (see red triangle in Fig. 1) with a constant concentration of 100, while the initial concentration across the domain is set to zero. We use the standard finite element method to obtain the numerical

- solutions of head and concentration. The numerical mesh adopted comprises  $41 \times 21 =$
- 861 nodes and 1,600 triangle elements. A uniform time step of 1 day is considered for
- a total simulation time of 10 days.
- The logarithm of conductivity  $(Y = \ln K)$  is considered as a spatially
- heterogeneous (correlated) Gaussian random field with an exponential covariance
- function ( $C_Y$ ) given by:

$$C_{\gamma} = \sigma_{\gamma}^{2} \exp\left(-\left(\frac{d_{x_{1}}}{\lambda_{x_{1}}} + \frac{d_{x_{2}}}{\lambda_{x_{2}}}\right)\right)$$
 (8)

- where  $\sigma_Y^2$  is the variance of Y;  $d_{x_i}$  (i = x, y) is separation (lag) distance between
- two given points in the *i*-direction;  $\lambda_{x_i}$  (with i = x, y) is the correlation length of Y in
- the *i*-direction. The corresponding mean of Y is denoted as  $\mu$ . The initial ensemble of
- Y fields is synthetically generated through the well-known and widely tested GSLIB
- suite (Deutsch and Journel, 1998) upon setting  $\lambda_{x_1}$  and  $\lambda_{x_2}$  equal to 1.0 and 0.5,
- respectively. The reference Y field (Fig. 1a) is generated upon setting  $\mu = 0.8$ ,  $\sigma_v^2$
- = 1.0,  $\lambda_{x_1}$  = 1.0, and  $\lambda_{x_2}$  = 0.5.
- The pumping rate (i.e.,  $q_s$ ),  $x_1$  and  $x_2$  -coordinates (denoted as  $x_{1,q_s}$  and
- $x_{2,q_s}$ , respectively) of the pumping location are considered to be random variables,
- ach associated by a Gaussian distribution. The gray zone in Fig. 1b encompasses the
- possible locations where a pumping well is operating. The initial collection (ensemble)
- of values of  $q_s$ ,  $x_{1,q_s}$ , and  $x_{2,q_s}$  and their reference counterparts are sampled from
- Gaussian distributions characterized by mean (standard deviation) equal to 0.50 (0.25),
- 1.00 (0.25), and 1.00 (0.25), respectively. These settings ensure that the randomly

generated samples of  $x_{1,q_s}$  and  $x_{2,q_s}$  are mostly within the coordinate ranges indicated by the gray zone in Fig. 2b. Reference values are  $q_s = 1.03$ ,  $x_{1,q_s} = 1.38$ , 372 and  $x_{2,q_s} = 1.40$  (see Fig. 1b, red cross symbol). Figure 1c depicts the simulated head 373 field associated with the reference conductivity field, pumping rate, and location. 374 375 Figure 1d depicts simulated concentrations at the final simulation time. Observations, including (steady-state) head and solute concentration at each time step, are collected 376 377 at a number (denoted as  $N_m$ ) of monitoring wells distributed across the aquifer 378 according to some pre-defined patterns (Fig. 1b-d). Each measurement is taken as the 379 sum of the simulated head (or concentration) and a white noise with zero mean and 380 standard deviation equal to  $\sigma_{obs}$ . To explore the potential of iES ROM, several showcases are designed to 381 highlight key features of interest. Four groups of test cases (TCs) are designed and 382 383 organized as detailed in the following (see also Table 1). **Group A.** It includes twelve TCs (i.e., TC1-TC12), enabling us to compare 384 performances of iES\_FSM and iES\_ROM associated with diverse values of 385 386 n when the pumping rate and locations are either known (TC1-TC6) or unknown (TC7-TC12). The dimension of the ROM is considered equal to {5, 387 10, 15, 20, 25, 30}, these values being consistent with those most commonly 388 analyzed in previous studies (Pasetto et al., 2014; Xia et al., 2020, 2025). 389 390 Group B. It includes four TCs (i.e., TC6 and TC13-TC15), enabling us to compare the performances of iES FSM and iES ROM with the largest 391 value of n analyzed (i.e, n = 30) and considering diverse values of  $N_{MC}$ 392

corresponding to  $\{30, 100, 500, 10,000\}$ . The latter are values of  $N_{MC}$ 393 394 commonly tested in previous studies (Chen and Zhang, 2006; Xia et al., 2021, 2024). 395 Group C. It includes five TCs (i.e., TC6 and TC16-TC19), designed to 396 397 analyze the ability of iES\_ROM to cope with diverse quality and quantity of available measurements. Performances of iES\_FSM and iES\_ROM are also 398 399 compared when  $\sigma_{obs} = \{0.001, 0.01, 0.1\}$  and the number of observation 400 locations corresponds to a value selected from {9 (Fig. 1b), 18 (Fig. 1c), 55 401 (Fig. 1d)}. Group D. It includes five TCs (i.e., TC6 and TC\_20-TC23), enabling us to 402 study the effect of  $\mu$  and  $\sigma_Y^2$  of the initial ensemble of Y on the 403 404 accuracies of estimates of conductivity and pumping rate and well location through iES\_FSM and iES\_ROM. Values of  $\mu$  and  $\sigma_{\gamma}^2$  of the initial 405 ensemble of Y fields are selected from  $\{-0.5, 1.2, 2.0\}$  and  $\{0.01, 1.0, 2.0\}$ , 406 respectively. 407 Group E. It includes six TCs (i.e., TC6 and TC24-TC28), with the aim of 408 409 investigating the effect of  $N_{sn}$  on the accuracies of the estimation of conductivity and well pumping rate and location through iES\_ROM and on 410 411 computation time requirements. Values of  $N_{sn}$  in TC24-TC28 and TC6 are 412 equal to 30, 100, 300, 500, 1,000, and 10,000, respectively. Note that, without specified otherwise, default settings for the above mentioned 413 414 TCs correspond to TC6 which is designed with n = 30,  $N_{MC} = 10,000$ ,  $N_{sn} = 10,000$ ,

- $N_m = 55$ ,  $\sigma_{obs} = 0.01$ , and values of  $\mu$  and  $\sigma_Y^2$  of the initial ensemble of Y equal
- to 1.2 and 1.0, respectively. Except for TC8-TC12, the source/sink term is associated
- with uncertainty.
- To quantify the accuracy of conductivity estimates through iES\_ROM and
- iES\_FSM, we consider absolute error between estimated and reference values of Y
- (denoted as  $E_Y$ ) and estimate of the standard deviation (denoted as  $S_Y$ ) which are
- defined as:

$$E_{Y} = \frac{1}{N} \sum_{i=1}^{N} \left| \left\langle Y_{i} \right\rangle^{est} - Y_{i}^{ref} \right|; \qquad S_{Y} = \sqrt{\frac{1}{N} \sum_{i=1}^{N} \left( \sigma_{Y,i}^{2} \right)^{est}}$$
 (9)

- where  $\left\langle Y_{i}\right\rangle ^{est}$ ,  $\left(\sigma_{Y,i}^{2}\right)^{est}$ , and  $Y_{i}^{ref}$  denote estimated (ensemble) mean and variance,
- and reference value of Y at the  $i^{th}$  cell of the numerical grid, respectively.
- Absolute errors and estimates of the standard deviations of  $\ln q_s$ ,  $x_{{\rm l},q_s}$ , and
- $x_{2,q_s}$  are employed to quantify the accuracy of the estimate of the pumping rate and
- well location:

$$E_{q_s} = \left| \left\langle \ln q_s \right\rangle^{est} - \ln q_s^{ref} \right|; \qquad E_{x_1} = \left| \left\langle x_{1,q_s} \right\rangle^{est} - x_{1,q_s}^{ref} \right|; \qquad E_{x_2} = \left| \left\langle x_{2,q_s} \right\rangle^{est} - x_{2,q_s}^{ref} \right|$$
 (10)

- where  $\langle \ln q_s \rangle^{est}$ ,  $\langle x_{1,q_s} \rangle^{est}$ , and  $\langle x_{2,q_s} \rangle^{est}$  indicate estimated (ensemble) mean values
- of  $\ln q_s$ ,  $x_{1,q_s}$ , and  $x_{2,q_s}$  respectively; and  $q_s^{ref}$ ,  $x_{1,q_s}^{ref}$ , and  $x_{2,q_s}^{ref}$  are the reference
- values of  $q_s$ ,  $x_{1,q_s}$ , and  $x_{2,q_s}$ , respectively. Estimates of the standard deviations of
- $\ln q_s$ ,  $x_{1,q_s}$ , and  $x_{2,q_s}$  are:

$$S_{q_s} = \sqrt{\left(\sigma_{\ln q_s}^2\right)^{est}}$$
;  $S_{x_1} = \sqrt{\left(\sigma_{x_1,q_s}^2\right)^{est}}$ ;  $S_{x_2} = \sqrt{\left(\sigma_{x_2,q_s}^2\right)^{est}}$  (11)

- where  $\left(\sigma_{\ln q_s}^2\right)^{est}$ ,  $\left(\sigma_{x_{1,q_s}}^2\right)^{est}$ , and  $\left(\sigma_{x_{2,q_s}}^2\right)^{est}$  denote estimated (ensemble) variances of
- $\ln q_s$ ,  $x_{1,q_s}$ , and  $x_{2,q_s}$ , respectively.

- As an additional metric, we then rely on the average absolute difference between
- available data and model results:

$$E_{obs} = \frac{1}{Q} \sum_{i=1}^{O} \left| \left\langle d_i \right\rangle^{up} - d_i^{ref} \right| \tag{12}$$

- where  $\langle d_i \rangle^{up}$  and  $d_i^{ref}$  correspond to the (updated) result of the simulation process
- and its reference observed counterpart at the  $i^{th}$  sampled location, respectively.

### 441 4. Results and discussion

### 4.1 Impact of the dimension of the reduced-order model (Group A)

- Figure 2 depicts  $E_Y$  (Fig. 2a),  $S_Y$  (Fig. 2b), and  $E_{obs}$  (Fig. 2c) versus the
- number of outer iterations for test cases (TCs) 1-6 obtained through iES\_ROM and
- iES\_FSM, when well pumping rate and location are uncertain. Note that results
- obtained through iES\_FSM are independent of n (and are identical among TCs 1-6)
- and are taken as references. Percentage differences (denoted as  $\Delta E_{\gamma}$ ) between the
- values of  $E_y$  obtained through iES\_ROM and iES\_FSM are depicted in Fig. 2d.
- Corresponding results associated with percentage differences between values of  $S_{\gamma}$
- $(\Delta S_{\gamma})$  and of  $E_{obs}$   $(\Delta E_{obs})$  are depicted in Fig. 2e and 2f, respectively.
- Values of  $E_{Y}$ ,  $S_{Y}$ , and  $E_{obs}$  obtained at the end of the iteration procedure
- through iES\_ROM generally decrease with n. When n = 25 or 30, the values of  $E_y$
- and  $S_y$  based on iES\_ROM tend to approach their counterparts obtained through
- iES FSM. The latter generally correspond to the lowest values across TCs 1-6. These
- findings are consistent with the observation (Xia et al., 2020; 2025) that accuracy of
- ROM for n = 30 and FSM are very similar for solute transport. They are also in line
- with the results of Li et al. (2013b), who documented a high degree of correlation

between simulated concentrations provided by their ROM and FSM for non-reactive 458 459 transport. As expected, the solution accuracy of ROM increases with n. When n approaches the dimension of the FSM (i.e., the representativeness of the basis 460 functions is essentially guaranteed), the ROM-based solution approaches its 461 FSM-based counterpart. 462 Figure 3 depicts  $E_Y$  (Fig. 3a),  $S_Y$  (Fig. 3b), and  $E_{obs}$  (Fig. 3c) for TCs 7-12 463 464 obtained through iES\_ROM and iES\_FSM when the well characteristics are 465 deterministically known. Similar to above, results obtained through iES\_FSM are identical among TCs 7-12 and are taken as reference. Values of  $\Delta E_{\gamma}$ ,  $\Delta S_{\gamma}$ , and 466  $\Delta E_{obs}$  are depicted in Fig. 3d, 3e, and 3f, respectively. 467 Consistent with what one can observe in Fig. 2, values of  $E_{\rm Y}$ ,  $S_{\rm Y}$ , and  $E_{\rm obs}$ 468 obtained at the end of the iteration procedure for TCs 7-12 through iES\_ROM 469 generally decrease with n. Except for the cases where n = 5 or 10 (corresponding to 470 low solution accuracy of ROM), values of  $E_{\rm Y}$ ,  $S_{\rm Y}$ , and  $E_{\rm obs}$  for TCs 9-12 based on 471 either iES\_ROM or iES\_FSM are lower than their counterparts related to TCs 3-6. 472 473 These results suggest that the accuracy of conductivity estimates is lower when  $q_s$  is uncertain compared to the case where  $q_s$  is deterministic. 474 Figure 4 depicts the values of  $E_{x_1}$  (Fig. 4a),  $E_{x_2}$  (Fig. 4b),  $E_{q_s}$  (Fig. 4c),  $S_{x_1}$ 475 (Fig. 4d),  $S_{x_2}$  (Fig. 4e), and  $S_{q_1}$  (Fig. 4f) versus the number of outer iterations for 476 TCs 1-6 obtained through iES\_ROM and iES\_FSM. Values of  $E_{x_1}$ ,  $E_{x_2}$ , and  $E_{q_s}$ 477 obtained through iES ROM approach their iES FSM-based counterparts as n 478 increases. This is consistent with the observation that increasing n improves the 479

accuracy of the ROM-based solution (see also Li et al., 2013b), therefore enhancing 480 481 the accuracy of the identification of the well attributes. Figure 5 depicts the estimated (ensemble) Y fields for TCs 1-6 obtained through 482 iES\_ROM and iES\_FSM, together with their reference Y field. The white circle and 483 cross symbols in Fig. 5 denote the estimated and reference locations of the pumping 484 well, respectively. As n increases, the estimated Y field obtained through iES\_ROM 485 486 (Fig. 5a-5f) approaches its iES\_FSM-based counterpart and the reference Y field (Fig. 487 5h). The accuracy of the iES\_ROM-based estimate of the location of the pumping well generally increases with n, consistent with the nature of the findings illustrated in 488 Fig. 4. Figure 6 depicts the estimated (ensemble) Y variance fields for TCs 1-6 based 489 on iES ROM and iES FSM. The white circle and cross symbols therein denote the 490 identified and reference locations of the pumping well, respectively. These results 491 492 show that the variance of Y is overestimated when n is small. This is related to the observation that small values of n correspond to large modeling errors (i.e., low 493 solution accuracy) of ROM (as also seen in Li et al. (2013b) and Pasetto et al. (2017)). 494 495 The latter, in turn, imprint the low accuracy of conductivity estimates (see Fig. 5a in the case of n = 5) and yield overestimated values for the variance of Y (see Fig. 6a). 496 497 Figure 7 depicts the empirical probability density function (PDF) of  $x_{1,q_s}$ ,  $x_{2,q_s}$ , and  $\ln q_s$  at the end of the iteration procedure for TCs 1, 2, 4, and 6 as obtained 498 499 through iES\_ROM and iES\_FSM, together with their counterparts associated with initial guess (black solid) and reference values (black dashed). One can observe that 500 large values of n yield high accuracy for  $x_{1,q_s}$  and  $x_{2,q_s}$  estimates, as visually 501

502 indicated by the compact supports associated with the empirical PDFs of  $x_{1,a}$  (Fig. 7a) and  $x_{2,q}$  (Fig. 7b). The accuracy of the estimate of  $q_s$  is already acceptable 503 when n = 5. 504 As an additional element, we explore the way the choice of the value of n505 506 impacts the local PDFs of hydraulic head and solute concentration. We do so upon considering the results associated with three reference points (i.e., I, II, and III in Fig. 507 508 1d) that are aligned in the direction of the mean groundwater flow. Figure 8 depicts 509 the (sample) PDFs of (hydraulic) head at these three selected locations (Figs. 8a-8c) 510 obtained through iES\_ROM and iES\_FSM at the end of the iteration procedure for TCs 1, 2, 4, and 6. Black solid lines included therein indicate reference head values. 511 Note that the PDFs stemming from iES FSM peak at values very close to their 512 513 reference counterparts. Hence, the corresponding empirical PDFs are considered as 514 reference. The logarithm absolute difference (ΔPDF, evaluated as the pointwise log-ratio of the densities and corresponding to a local measure of relative likelihood 515 between two empirical PDFs) between the PDFs of the head at points I-III obtained 516 517 through iES\_ROM based on diverse values of n and their counterpart based on iES\_FSM are also shown in Figs. 8d-8f, respectively. One can see that a large value of 518 n (e.g., n = 30 for TC6) corresponds to high accuracy of the PDF of head, as 519 quantified through a low value of  $\Delta PDF$ . Although the head solution is obtained by 520 521 solving FSM, the accuracy of the conductivity estimate is impacted by n. The latter, 522 therefore, impacts the accuracy of heads. Fig. 9 depicts results related to solute concentration. As expected, the PDFs stemming from iES FSM peak at values very 523

close to their reference counterparts also in this case. Consistent with Fig. 8, a large 524 525 value of n (e.g., 30 for TC6) corresponds to high accuracy in the delineation of the PDF of solute concentration. 526 As a complement to these results, values of the Kullback-Leibler Divergence 527 528 (KLD) between the (sample) PDFs of head at the three reference points at the last outer iteration obtained through iES\_FSM ( $h_{FSM}$ ) and iES\_ROM ( $h_{ROM}$ ) with n = 5529 530 (TC1), 10 (TC2), 20 (TC4), and 30 (TC6) are listed in Table S1 (see supplementary 531 information). We recall that values of  $KLD(h_{ROM}||h_{FSM})$  (or  $(KLD(h_{FSM}||h_{ROM}))$ 532 quantify (in a global sense) information loss when using  $h_{FSM}$  ( $h_{ROM}$ ) to approximate  $h_{\text{ROM}}$  ( $h_{\text{FSM}}$ ). Values of KLD( $h_{\text{ROM}} || h_{\text{FSM}}$ ) generally increase with n. This indicates that 533 the difference between PDFs of  $h_{ROM}$  and  $h_{FSM}$  decrease as n increases. While the 534 highest values of KLD( $h_{FSM}||h_{ROM}$ ) correspond to n=5, no clear decreasing trends 535 536 with increasing n are observed. Furthermore, the difference between  $KLD(h_{ROM}||h_{FSM})$ and  $KLD(h_{FSM}||h_{ROM})$  generally decreases as n increases. This is related to the 537 observation that the accuracy of ROM tends to increase as the dimension of the 538 539 reduced-order model increase. Values of KLD between the empirical PDFs of solute concentrations at the three selected reference points at the last outer iteration obtained 540 through iES\_FSM ( $c_{\text{FSM}}$ ) and iES\_ROM ( $c_{\text{ROM}}$ ) with n = 5 (TC1), 10 (TC2), 20 (TC4), 541 and 30 (TC6) are listed in Table S2 (see supplementary information). 542 543 **4.2** Effect of the ensemble size (Group B) Figure 10 depicts iES\_ROM- and iES\_FSM-based values of  $E_v$  (Fig. 10a),  $S_v$ 544 (Fig. 10b), and  $E_{obs}$  (Fig. 10c) versus the number of outer iterations for TCs 6 and 545

13-15. Values of  $E_{\rm Y}$  and  $E_{\rm obs}$  decrease as the ensemble size  $N_{\rm MC}$  increases (while 546 the value of  $S_y$  increases) regardless of the approach employed. With reference to 547 TC13, we note that when  $N_{MC} = 30$  the values of  $E_y$  decrease during the course of 548 549 the first outer iterations to then increase during the last outer iterations, values of  $S_{\gamma}$ 550 dropping rapidly during the iteration procedure, regardless of the approach employed. This phenomenon is typically linked to the occurrence of filter inbreeding caused by a 551 limited ensemble size (Chen and Zhang, 2006; Xia et al., 2018; 2024). Values of  $E_{\gamma}$ 552 and  $S_{\gamma}$  for TCs 6 and 13-15 obtained through iES\_ROM are overall similar to those 553 associated with iES\_FSM. The iES\_ROM-based value of  $E_{obs}$  obtained at the end of 554 the iteration procedure for a given TC is typically larger than its iES\_FSM-based 555 counterpart. This is linked to the observation that the limited system dimension of 556 557 ROM induces low accuracy of concentrations and (possibly) heads due to low 558 accuracy of conductivity estimates, pumping rate, and well locations. Figure 11 depicts the values of  $E_{x_1}$  (Fig. 11a),  $E_{x_2}$  (Fig. 11b),  $E_{q_s}$  (Fig. 11c), 559 560  $S_{x_1}$  (Fig. 11d),  $S_{x_2}$  (Fig. 11e), and  $S_{q_2}$  (Fig. 11f) versus the number of outer iterations for TCs 6 and 13-15 obtained through iES\_ROM and iES\_FSM. When 561 562 increasing  $N_{MC}$ , values of  $E_{x_1}$ ,  $E_{x_2}$ , and  $E_{q_3}$  obtained through either iES\_ROM or iES\_FSM do not show a clear trend. Values of  $S_{x_1}$ ,  $S_{x_2}$ , and  $S_{q_s}$  generally increase 563 with  $N_{MC}$ , a result that is consistent with the findings encapsulated in Fig. 10b. Similar 564 565 findings are also documented by Xu and Gómez-Hernández (2018, their Fig. 17), who show that, when considering joint identification of contaminant sources and hydraulic 566 conductivities, the accuracy of estimates of key attributes characterizing contaminant 567

sources does not necessarily improve after some time and as data assimilation 568 569 progresses. We further note that jointly estimating conductivity and identifying source/sink term attributes (in terms of flow rate and location) is associated with a 570 highly nonlinear optimization process. Hence, the accuracies of location and pumping 571 572 rate estimation through iES\_FSM are not always higher than those stemming from 573 iES\_ROM in terms of the values of the metrics employed (i.e.,  $E_{x_1}$ ,  $E_{x_2}$ , and  $E_{q_3}$ ). 574 Figures 12 depicts the estimated (ensemble mean) Y fields for TCs 6 and 13-15 575 obtained through iES\_ROM and iES\_FSM. Figure 13 depicts the associated Y 576 variance fields for TCs 6 and 13-15 obtained through iES ROM and iES FSM. The white (black) circle and cross symbols in Fig. 12 (or Fig. 13) represent the identified 577 and the reference locations of the pumping well, respectively. Visual comparison of 578 579 Fig. 12 and Fig. 5h suggests that the estimated Y fields rendered through an ensemble 580 size  $N_{MC} = 100$  (i.e., TC14) obtained through iES\_ROM and iES\_FSM are the closest ones to the reference Y field. Nevertheless, jointly analyzing Figs. 10a, 12, and 13 581 reveal that the estimated Y field corresponding to  $N_{MC} = 10,000$  (TC6) obtained 582 583 through iES\_ROM is the one most closest to the reference Y field in terms of  $E_Y$  (= 0.41). Additionally, the identified and reference locations of the pumping well 584 obtained through either iES\_ROM or iES\_FSM are close to each other, thus 585 supporting the capability of both approaches to identify the well location. 586 587 4.3 Effect of quality and available number of observations (Group C) Table 2 lists values of  $E_Y$ ,  $S_Y$ ,  $E_{obs}$ ,  $E_{x_1}$ ,  $E_{x_2}$ ,  $E_{q_s}$ ,  $S_{x_1}$ ,  $S_{x_2}$ , and  $S_{q_s}$  at the 588

end of iteration procedure for TC16 (characterized by  $\sigma_{obs} = 0.001$ ), TC6 ( $\sigma_{obs} = 0.001$ )

| 590 | 0.01), and TC17 ( $\sigma_{obs} = 0.1$ ) obtained through iES_ROM and iES_FSM. Values of                                      |
|-----|-------------------------------------------------------------------------------------------------------------------------------|
| 591 | $E_{\rm Y}$ , $S_{\rm Y}$ , $E_{obs}$ , $E_{\rm x_1}$ , and $E_{q_{\rm s}}$ generally increase as the quality of observations |
| 592 | deteriorates, i.e., $\sigma_{obs}$ increasing from 0.001 to 0.1. These results are also consistent                            |
| 593 | with prior findings by Xia et al. (2018) according to which accuracy of conductivity                                          |
| 594 | estimates increases as the quality of observations improves. Values of $E_{x_2}$ obtained                                     |
| 595 | through iES_ROM and iES_FSM do not monotonically decrease as $\sigma_{\scriptscriptstyle obs}$ decreases.                     |
| 596 | This is typically related to the strong nonlinear nature associated with the optimization                                     |
| 597 | process (see also Xu and Gómez-Hernández, 2018).                                                                              |
| 598 | Figure 14 depicts iES_ROM- and iES_FSM-based values of $E_{\gamma}$ (Fig. 14a), $S_{\gamma}$                                  |
| 599 | (Fig. 14b), and $E_{\it obs}$ (Fig. 14c) versus the number of outer iterations for TCs 6 and                                  |
| 600 | 18-19. Values of $E_Y$ (or $S_Y$ ) for TCs 18 (where the number of monitoring wells is                                        |
| 601 | $N_m = 9$ ), 19 ( $N_m = 18$ ), and 6 ( $N_m = 55$ ) obtained through iES_ROM are similar to                                  |
| 602 | their iES_FSM-based counterparts and decrease as $N_{\scriptscriptstyle m}$ increases. Values of $E_{\scriptscriptstyle obs}$ |
| 603 | obtained through iES_FSM decrease as $N_{\scriptscriptstyle m}$ increases, while iES_ROM-based                                |
| 604 | results do not display a clear trend with $N_m$ . This result may be attributed to the fact                                   |
| 605 | that, while increasing the number of monitoring wells enhances the amount of                                                  |
| 606 | information available for estimating hydraulic conductivity, errors introduced through                                        |
| 607 | model reduction influence the evolution of the solute concentration mismatch between                                          |
| 608 | observations and simulations during the iterative calibration process.                                                        |
| 609 | Figure 15 depicts the values of $E_{x_1}$ (Fig. 15a), $E_{x_2}$ (Fig. 15b), $E_{q_s}$ (Fig. 15c),                             |
| 610 | $S_{x_1}$ (Fig. 15d), $S_{x_2}$ (Fig. 15e), and $S_{q_s}$ (Fig. 15f) versus the number of outer                               |
| 611 | iterations for TCs 6 and 18-19 obtained through iES_ROM and iES_FSM. Values of                                                |

 $E_{x_1}$  (  $E_{x_2}$  ,  $E_{q_s}$  ,  $S_{x_1}$  ,  $S_{x_2}$  ,  $S_{q_s}$  , or  $S_Y$  ) for TCs 18 (for a number  $N_m=9$  of monitoring wells), 19 ( $N_m = 18$ ), and 6 ( $N_m = 55$ ) obtained through either 613 iES\_ROM or iES\_FSM decrease as  $N_m$  increases. Values of the same metric (i.e., 614 615  $E_{x_1}$ ,  $E_{x_2}$ ,  $E_{q_3}$ ,  $S_{x_1}$ ,  $S_{x_2}$ , or  $S_{q_3}$ ) obtained through iES\_ROM and iES\_FSM are 616 overall close to each other. Figure 16 depicts the empirical PDF of  $x_{1,q_s}$ ,  $x_{2,q_s}$ , and  $\ln q_s$  at the end of the 617 618 iteration procedure for TCs 18, 19, and 6 obtained through iES\_ROM and iES\_FSM, 619 together with their reference counterparts (black dashed lines). One can observe that increasing  $N_m$  leads to improved accuracy of the identification of pumping well 620 attributes, as suggested by the reduced support and location of the peaks of the PDFs 621 of  $x_{1,q_s}$  (Fig. 16a),  $x_{2,q_s}$  (Fig. 16b), and  $\ln q_s$  (Fig. 16c) obtained through either 622 623 iES\_ROM or iES\_FSM and observed as  $N_m$  varies from 9 to 55. On the basis of 624 these results, it is hard to tell which approach provides higher accuracy of pumping well identification, solely in terms of Fig. 16. To complement these findings, Table S3 625 (see supplementary information) lists the values of KLD between the empirical PDFs 626 of  $x_{1,q_s}$  (  $x_{2,q_s}$  , or  $\ln q_s$  ) obtained through iES\_FSM (denoted as  $p_{\rm FSM}$ ) and 627 iES\_ROM (denoted as  $p_{ROM}$ ) with n = 30, considering  $N_m = 9$  (TC18), 18 (TC19), and 628 55 (TC6), respectively. Values of KLD( $p_{j\text{ROM}}||p_{j\text{FSM}}$ ) (with  $j=x_{1,q_s}$ ,  $x_{2,q_s}$ , and  $\ln q_s$ 629 630 show an overall decreasing trend as  $N_m$  increase, while  $\text{KLD}(p_{j\text{FSM}}||p_{j\text{ROM}})$ consistently decreases with  $N_m$ . These results are consistent with the observation that 631 increasing the number of monitoring wells improves the accuracy of conductivity 632 estimates (as also seen by Tong et al. (2010) and Xia et al. (2018)) as well as pumping 633

between the corresponding PDFs. 635 **4.4** Effect of the mean and variance of the initial ensemble of *Y* (Group **D**) 636 637 Table 3 lists the values of  $E_y$ ,  $S_y$ ,  $E_{obs}$ ,  $E_{x_1}$ ,  $E_{x_2}$ ,  $E_{q_2}$ ,  $S_{x_1}$ ,  $S_{x_2}$ , and  $S_{q_2}$  at the end of the iteration procedure for TCs 20 (characterized by a mean  $\mu$  = -0.5 of 638 639 the initial ensemble of Y), 6 ( $\mu = 1.2$ ), and 21 ( $\mu = 2.0$ ) obtained through 640 iES\_ROM and iES\_FSM. We recall that the mean value employed to generate the reference Y field is equal to 0.8. When the discrepancy between  $\mu$  and the mean 641 642 value of the reference Y field increases, the error metrics employed display an overall increase,  $E_{x_1}$  and  $E_{q_s}$  constituting notable exceptions. This finding is consistent 643 with the behavior documented by Xia et al. (2024) who considered two 644 645 correlation-based localization approaches to assess conductivity estimation accuracy 646 with respect to the mean of the initial ensemble of Y. Overall, this result is related to the highly nonlinear nature of the optimization process (see also Sections 4.2 and 4.3) 647 stemming from the joint estimation of conductivities and pumping well attributes. 648 Table 4 lists the values of  $E_Y$ ,  $S_Y$ ,  $E_{obs}$ ,  $E_{x_1}$ ,  $E_{x_2}$ ,  $E_{q_x}$ ,  $S_{x_1}$ ,  $S_{x_2}$ , and  $S_{q_x}$  at 649 the end of the iteration procedure for TCs 22 (characterized by a variance  $\sigma_Y^2 = 0.01$ 650 of the initial ensemble of Y), 6 ( $\sigma_{\rm Y}^2=1.0$ ), and 23 ( $\sigma_{\rm Y}^2=2.0$ ) obtained through 651 iES\_ROM and iES\_FSM. We recall that the reference Y field is characterized by a 652 unit variance. The values of  $E_{Y}$  and  $E_{x_2}$  obtained through both approaches increase 653 as the discrepancy between  $\sigma_Y^2$  and the variance of the reference Y field increases. 654 The values of  $E_{obs}$ ,  $E_{x_1}$ , and  $E_{q_s}$  obtained through both approaches generally 655

rate and well location through both approaches, thus, in turn, reducing discrepancies

increase with  $\sigma_{\gamma}^2$ . Similarly, values of metrics employed to quantify variability of the final ensemble of realizations (i.e.,  $S_Y$ ,  $S_{x_1}$ ,  $S_{x_2}$ , and  $S_{q_s}$ ) consistently increase 657 with  $\sigma_{v}^{2}$ . 658 A joint analysis of the results illustrated in Sections 4.1, 4.2, and 4.3 suggests 659 that  $E_{\gamma}$  and  $S_{\gamma}$  provided by both approaches show a consistent behavior as a 660 function of the key feature of interest. Otherwise, the response of the metrics 661 662 associated with the pumping well attributes provided by both approaches reflects the 663 enhanced nonlinearity of the associated optimization process. Additionally, the 664 accuracy of the conductivity estimate possibly contributes more to the minimization of the objective function than that of pumping well identification. Additionally, the 665 values of the metrics in Sections 4.1, 4.2, and 4.3 provided by the two approaches are 666 generally consistent with each other, thus supporting the representativeness of the 667 668 iES ROM-based results. 669

### 4.5 Effect of the snapshot size (Group E)

Table 5 lists percentage differences of the values of the performance metrics considered (i.e.,  $E_{Y}$ ,  $S_{Y}$ ,  $E_{obs}$ ,  $E_{x_1}$ ,  $E_{x_2}$ ,  $E_{q_s}$ ,  $S_{x_1}$ ,  $S_{x_2}$ , and  $S_{q_s}$ ) at the end of the iteration procedure for TCs 24-28 obtained through iES\_ROM, considering their counterparts through TC6 as references. These results show that the values of  $E_{\gamma}$ and  $S_Y$  systematically decrease as  $N_{sn}$  increases from 30 to 1,000, while the other metrics display an overall decreasing pattern. This is related to the observation that a larger snapshot size corresponds to a higher accuracy of basis functions (Pasetto et al., 2014). Otherwise, it is worth noting that snapshots are evaluated only once throughout

the entire data assimilation processes, thus resulting in a limited computational cost.

The CPU required time for running TC28 is about 13 minutes (using a processor 13th Gen Intel(R) Core(TM) i7-13700K 3.40 GHz with 32 GB RAM). CPU times for running TC6 through iES\_ROM and iES\_FSM are about 28 and 122 minutes, respectively. The CPU time to complete TC6 upon relying on iES\_FSM is about 9 times the CPU time for TC28 through iES\_ROM, while percentage differences associated with  $E_{\gamma}$  and  $S_{\gamma}$  are 0.50% and 0.21%, respectively. CPU time savings can become more pronounced during data assimilation for a groundwater system of large size, due to the higher memory requirements of iES\_FSM for storing and computing large-dimensional vectors and matrices as compared to iES\_ROM. These results support the ability of iES\_ROM to estimate conductivity and identify the uncertain features of a pumping well under the conditions analyzed.

### 5. Conclusions

This study addressed the joint estimation of (uncertain, spatially heterogeneous) hydraulic conductivities and attributes (location and flow rate) of a pumping well in a two-dimensional confined aquifer in the presence of (non-reactive) solute transport taking place across a steady-state flow field. Our analyses rest on an iterative Ensemble Smoother (iES) coupled with a Reduced-Order Model (ROM) for solute transport (the overall strategy being denoted as iES\_ROM). The ROM is constructed via Proper Orthogonal Decomposition (POD), using basis functions derived from the numerical solutions of the Full System Model (FSM) over the entire simulation period. The pumping well is characterized by its spatial coordinates ( $x_{1,q_s}$ ,  $x_{2,q_s}$ ) and a

700 constant pumping rate  $q_s$ . The ROM can achieve a solution accuracy similar to that of the FSM, while significantly reducing computational demands. Notably, as stated 701 above, the basis functions are computed only once throughout the iES ROM iteration 702 process, thus further enhancing efficiency. As a benchmark, the traditional iES 703 704 approach relying on the FSM (termed iES\_FSM) is also implemented to estimate conductivity and identify well attributes. 705 706 To assess the performance and robustness of the proposed iES\_ROM approach, 707 twenty-eight test cases (TCs 1-28) are designed and structured according to five 708 categories (Groups A-E; Section 3), each targeting different influencing factors. These 709 include the dimension of the reduced-order model (n), ensemble size ( $N_{mc}$ ), standard deviation of the white noise representing measurement error ( $\sigma_{obs}$ ), number of 710 711 monitoring wells  $(N_m)$ , mean  $(\mu)$  and variance  $(\sigma_v^2)$  of the initial log-conductivity field, and snapshot size  $(N_{sn})$ . The performance of iES\_ROM is systematically 712 compared with that of iES FSM using nine evaluation metrics, encompassing the 713 714 absolute error ( $E_y$ ; Equation (9)) and estimated standard deviation ( $S_y$ ; Equation (9)) between estimated and reference values of Y; the absolute errors and estimated 715 standard deviations of the pumping well coordinates and rate (Equation (10)); and the 716 717 average absolute difference between simulated and reference observations ( $E_{obs}$ ; Equation (12)). 718 719 Our work leads to the following major conclusions. 720 Both iES ROM and iES FSM yield accurate estimates of hydraulic

conductivity distributions and identify the pumping well attributes across a

wide range of tested conditions, including variations in model dimension, 722 ensemble size, measurement noise, number of monitoring wells, and 723 statistical properties of the initial ensemble. 724 The iES\_ROM approach achieves estimation accuracy similar to that of 725 726 iES\_FSM when using a moderate reduced-order dimension (n = 25 or 30). Otherwise, relying on a small dimension (e.g., n = 5) yields filter divergence 727 728 due to unaccounted model errors. Increasing n effectively mitigates this 729 issue and enhances the stability of the iES\_ROM performance. 730 When hydraulic conductivity and pumping well attributes are jointly estimated, both iES\_ROM and iES\_FSM exhibit a slight reduction in the 731 accuracy of conductivity estimates compared to scenarios where only 732 733 conductivity is estimated. This trend is reflected in the values of  $E_y$ ,  $S_y$ , and  $E_{obs}$  across TCs 1-12. Under such joint estimation, results in terms of 734  $E_{\rm Y}$ ,  $S_{\rm Y}$ , and  $E_{\rm obs}$  with respect to different influencing factors remain of 735 acceptable quality for both iES\_ROM and iES\_FSM, consistent with the 736 737 patterns observed in conductivity-only estimation. The behaviors of the remaining performance metrics are mutually consistent and within 738 acceptable ranges, although somewhat less orderly. 739 Relying on the iES\_ROM approach yields an accuracy similar to that of 740 741 iES\_FSM in estimating hydraulic conductivity and identifying pumping well attributes for both moderate ( $N_{sn} = 500$  or 1000) and large ( $N_{sn} = 10000$ ) 742 ensemble sizes. This result supports its robustness with respect to ensemble 743

size selection. 744 745 In terms of computational efficiency, iES\_ROM yields substantial time savings compared to iES\_FSM. For instance, with  $N_{sn} = 500$  and n = 30, 746 the CPU times for iES\_ROM and iES\_FSM are approximately 28 and 122 747 748 minutes, respectively (i.e., iES\_FSM requires a computation time that is about nine times longer while yielding similar estimation accuracy). 749 750 Author contributions. All authors contributed to the preparation of the manuscript. 751 Acknowledgments. This work was supported by the National Nature Science 752 Foundation of China (Grant No. 42002247), Nature Science Foundation of Fujian Province, China (Grant No. 2025J01529; 2025J08248), and Opening Fund of Key 753 Laboratory of Geohazard Prevention of Hilly Mountains, Ministry of Natural 754 755 Resources (FJKLGH2024K008). M.R. acknowledges funding from the National 756 Recovery and Resilience Plan (NRRP), mission 4 component 2 investment 1.4 - call for tender no. 3138 of 16 December 2021, rectified by decree no. 3175 of 18 757 December 2021 of Italian Ministry of University and Research funded by the 758 759 European Union - NextGenerationEU, project code CN\_00000033, concession decree no. 1034 of 17 June 2022 adopted by the Italian Ministry of University and Research, 760 CUP D43C22001250001, project title "National Biodiversity Future Center - NBFC". 761 References 762 763 Asher, M.J., Croke, B.F.W., Jakeman, A.J., Peeters, L.J.M., 2015. A review of surrogate models and their application to groundwater modeling. Water Resour. 764 Res. 51, 5957–5973. 765

- Ballio, F., Guadagnini, A., 2004. Convergence assessment of numerical Monte Carlo
- simulations in groundwater hydrology. Water Resour. Res. 40, W04603.
- Boyce, S.E., Nishikawa, T., Yeh, W.W.G., 2015. Reduced order modeling of the
- newton formulation of modflow to solve unconfined groundwater flow. Adv.
- Water Resour. 83, 250–262.
- Chen, Y., Zhang, D., 2006. Data assimilation for transient flow in geologic formations
- via ensemble Kalman filter. Adv Water Resour., 29(8): 1107-22.
- Chen, Y., Oliver, D. S., 2013. Levenberg–Marquardt forms of the iterative ensemble
- 5774 smoother for efficient history matching and uncertainty quantification.
- Computational Geosciences, 17(4): 689-703.
- Chen, Z., Jaime Gomez-Hernandez, J., Xu, T., Zanini, A., 2018. Joint identification of
- contaminant source and aquifer geometry in a sandbox experiment with the
- restart ensemble kalman filter. J. Hydrol. 564, 1074–1084.
- Deutsch, C.V., Journel, A.G., 1998. GSLIB: Geostatistical Software Library and
- User's Guide, second ed. Oxford University Press, New York.
- Evensen, G., 2009. Data Assimilation: The Ensemble Kalman Filter. Data
- Assimilation: The Ensemble Kalman Filter.
- Ju, L., Zhang, J., Meng, L., et al. 2018. An adaptive Gaussian process-based iterative
- ensemble smoother for data assimilation. Adv. Water Resour., 115: 125-35.
- Li, X., Chen, X., Hu, B.X., Navon, I.M., 2013a. Model reduction of a coupled
- numerical model using proper orthogonal decomposition. J. Hydrol. 507,
- 227–240.

Li, X., Hu, B. X., 2013b. Proper orthogonal decomposition reduced model for mass 788 789 transport in heterogeneous media. Stochastic Environmental Research and Risk Assessment, 27(5): 1181-91. 790 Luo, Z., Li, H., Zhou, Y., et al. 2012. A reduced finite element formulation based on 791 792 POD method for two-dimensional solute transport problems. Journal of Mathematical Analysis and Applications, 385(1): 371-83. 793 794 Luo, X., Bhakta, T., 2020. Automatic and adaptive localization for ensemble-based 795 history matching. Journal of Petroleum Science and Engineering, 184: 106559. 796 Mo, S., Zabaras, N., Shi, X., et al. 2019. Deep Autoregressive Neural Networks for High-Dimensional Inverse Problems in Groundwater Contaminant Source 797 Identification. Water Resour. Res., 55(5): 3856-81. 798 799 Pasetto, D., Guadagnini, A., Putti, M., 2011. POD-based Monte Carlo approach for 800 the solution of regional scale groundwater flow driven by randomly distributed Adv. Water 34(11): 1450-1463. recharge. Resour., 801 DOI:10.1016/j.advwatres.2011.07.003 802 803 Pasetto, D., Putti, M., Yeh, W.W.G., 2013. A reduced-order model for groundwater flow equation with random hydraulic conductivity: Application to Monte Carlo 804 methods. Water Resour. Res., 49(6): 3215-3228. DOI:10.1002/wrcr.20136 805 Pasetto, D., Guadagnini, A., Putti, M., 2014. A reduced-order model for Monte Carlo 806 807 simulations of stochastic groundwater flow. Computational Geosciences, 18(2): 157-169. DOI:10.1007/s10596-013-9389-4 808 Pasetto, D., Ferronato, M., Putti, M., 2017. A reduced order model-based

| 810 | preconditioner for the efficient solution of transient diffusion equations.           |
|-----|---------------------------------------------------------------------------------------|
| 811 | International Journal for Numerical Methods in Engineering, 109(8): 1159-1179.        |
| 812 | DOI:10.1002/nme.5320                                                                  |
| 813 | Pinnau, R., 2008. Model Reduction via Proper Orthogonal Decomposition /Schilders,     |
| 814 | W. H. A., Van Der Vorst, H. A., Rommes, J. Model Order Reduction: Theory,             |
| 815 | Research Aspects and Applications. Berlin, Heidelberg; Springer Berlin                |
| 816 | Heidelberg. 95-109.                                                                   |
| 817 | Razavi, S., Tolson, B.A., Burn, D.H., 2012. Review of surrogate modeling in water     |
| 818 | resources. Water Resour. Res. 48                                                      |
| 819 | Rizzo, C., de Barros, F., Perotto, S., Oldani, L., Guadagnini, A., 2018. Adaptive POD |
| 820 | model reduction for solute transport in heterogeneous porous media. Computat.         |
| 821 | Geosci. 22, 297–308.                                                                  |
| 822 | Stanko, Z.P., Boyce, S.E., Yeh, W.WG., 2016. Nonlinear model reduction of             |
| 823 | unconfined groundwater flow using pod and deim. Adv. Water Resour. 97,                |
| 824 | 130–143.                                                                              |
| 825 | Tong, J., Hu, B. X., Yang, J., 2010. Using data assimilation method to calibrate a    |
| 826 | heterogeneous conductivity field conditioning on transient flow test data.            |
| 827 | Stochastic Environmental Research and Risk Assessment, 24(8): 1211-23.                |
| 828 | Xia, CA., Luo, X., Hu, B.X., Riva, M., Guadagnini, A., 2021. Data assimilation with   |
| 829 | multiple types of observation boreholes via the ensemble Kalman filter                |
| 830 | embedded within stochastic moment equations. Hydrol. Earth Syst. Sci., 25(4):         |
| 831 | 1689-1709. DOI:10.5194/hess-25-1689-2021                                              |

- Xia, C.-A., Pasetto, D., Hu, B.X., Putti, M., Guadagnini, A., 2020. Integration of
- moment equations in a reduced-order modeling strategy for Monte Carlo
- simulations of groundwater flow. J. Hydrol., 590: 125257.
- DOI:https://doi.org/10.1016/j.jhydrol.2020.125257
- Xia, C.-A., Guadagnini, A., Hu, B. X., Riva, M., Ackerer, P., 2019. Grid convergence
- for numerical solutions of stochastic moment equations of groundwater flow,
- Stoch. Environ. Res. Risk Assess., 33 (8-9), 1565-1579,
- https://doi.org/10.1007/s00477-019-01719-6.
- Xia, C.-A., Hu, B.X., Tong, J., Guadagnini, A., 2018. Data Assimilation in
- Density-Dependent Subsurface Flows via Localized Iterative Ensemble Kalman
- Filter. Water Resour. Res., 54(9): 6259-6281. DOI:10.1029/2017wr022369
- Xia, C.-A., Li, J., Riva, M., et al., 2024. Characterization of conductivity fields
- through iterative ensemble smoother and improved correlation-based adaptive
- localization. J. Hydrol., 634: 131054.
- Xia, C.-A., Wang, H., Jian, W., et al., 2025. Reduced-order Monte Carlo simulation
- framework for groundwater flow in randomly heterogeneous composite
- transmissivity fields. J. Hydrol., 651: 132593.
- Xu, T., Jaime Gomez-Hernandez, J., 2018. Simultaneous identification of a
- contaminant source and hydraulic conductivity via the restart normal-score
- ensemble Kalman filter. Adv. Water Resour., 112: 106-123.
- B52 DOI:10.1016/j.advwatres.2017.12.011
- Zhang, D., 2002. Stochastic Method for Flow in Porous Media Coping with

## https://doi.org/10.5194/egusphere-2025-5320 Preprint. Discussion started: 7 November 2025 © Author(s) 2025. CC BY 4.0 License.

| 854 | Uncertainties. Academic Press, Sand Diego, California.                         |
|-----|--------------------------------------------------------------------------------|
| 855 | Zhang J, Lin G, Li W, et al. An Iterative Local Updating Ensemble Smoother for |
| 856 | Estimation and Uncertainty Assessment of Hydrologic Model Parameters With      |
| 857 | Multimodal Distributions. Water Resour Res, 2018, 54(3): 1716-33.              |
| 858 |                                                                                |

Fig. 1 (a) Reference field of  $Y = \ln K$ ; (b) boundary conditions for groundwater flow an solute transport together with spatial distribution of 9 monitoring wells and reference location for the pumping well (shaded gray area corresponds to the region comprising the unknown location of the well); (c) hydraulic head corresponding to the reference Y field; and (d) solute concentration corresponding the reference Y field at final time step, including three selected locations (i.e., I, II, and III) at which empirical probability density functions of solute concentration is computed and considered for illustration purposes. Circles in (b), (c) and (d) correspond to the location of the 9, 18 and 55 monitoring wells, respectively, employed in the study (see Section 3 and Table 1).

Fig. 2 Values of (a)  $E_{\gamma}$ , (b)  $S_{\gamma}$ , and (c)  $E_{obs}$  versus the number of outer iterations obtained through iES\_ROM considering various dimensions of reduced-order model (with  $n=5,\ 10,\ 15,\ 20,\ 25,\$ and 30 for TCs 1-6, respectively) and iES\_FSM (which provides identical results for TCs 1-6) for ensemble size  $N_{MC}=10,000;$  corresponding percentage differences between the values of (d)  $E_{\gamma}$  ( $\Delta E_{\gamma}$ ), (e)  $S_{\gamma}$  ( $\Delta S_{\gamma}$ ), and (f)  $E_{obs}$  ( $\Delta E_{obs}$ ) evaluated through iES\_ROM and iES\_FSM.

Fig. 3 Values of (a)  $E_{\gamma}$ , (b)  $S_{\gamma}$ , and (c)  $E_{obs}$  versus the number of outer iterations

obtained through iES\_ROM considering various dimensions of reduced-order model (with n = 5, 10, 15, 20, 25, and 30 for TCs 7-12, respectively) and iES\_FSM (which provides identical results for TCs 7-12), when the pumping rate and location are previously known and for an ensemble size  $N_{MC} = 10,000$ ; corresponding percentage

differences between the values of (d)  $E_y$  ( $\Delta E_y$ ), (e)  $S_y$  ( $\Delta S_y$ ), and (f)  $E_{obs}$ 

 $(\Delta E_{obs})$  evaluated through iES\_ROM and iES\_FSM.

896 897

889

890 891

892

893

894

895

.

Fig. 4 Values of (a)  $E_{x_1}$ , (b)  $E_{x_2}$ , (c)  $E_{q_s}$ , (d)  $S_{x_1}$ , (e)  $S_{x_2}$ , and (f)  $S_{q_s}$  versus the number of outer iterations obtained through iES\_ROM considering various dimensions of reduced-order model (with n=5, 10, 15, 20, 25, and 30 for TCs 1-6, respectively) and iES\_FSM (which provides identical results for TCs 1-6), when  $N_{MC}=10,000$ .

Fig. 5 Estimated (ensemble) mean Y fields at the final outer iteration through iES\_ROM considering different n (equal to (a) 5, (b) 10, (c) 15, (d) 20, (e) 25, and (f) 30 for TCs 1-6, respectively) and (g) iES\_FSM (which provides identical results for TCs 1-6) when  $N_{MC} = 10,000$ ; (h) reference Y field.

Fig. 6 Estimated (ensemble) Y variance fields at the final outer iteration through iES\_ROM considering different n (equal to (a) 5, (b) 10, (c) 15, (d) 20, (e) 25, and (f) 30 for TCs 1-6, respectively) and (g) iES\_FSM (which provides identical results for TCs 1-6), when  $N_{MC} = 10,000$ .

Fig. 7 Empirical PDFs of (a)  $x_{1,q_s}$ , (b)  $x_{2,q_s}$ , and (c)  $\ln q_s$  at the final outer iteration through iES\_ROM considering various values of n (equal to 5, 10, 20, and 30 for TCs 1, 2, 4, and 6, respectively) and iES\_FSM (which provides identical results for TCs 1, 2, 4, and 6) when  $N_{MC}=10,000$ ; corresponding reference values are indicated by black dashed lines.

Fig. 8 Empirical PDFs of hydraulic head at points (a) I, (b) II, and (c) III (see Fig. 1) at the final outer iteration obtained through iES\_ROM considering different values of n (equal to 5 (cyan dashed curve), 10 (magenta), 20 (green), and 30 (blue) for TCs 1, 2, 4, and 6, respectively) and iES\_FSM (red solid curve; identical results for TCs 1, 2, 4, and 6) when  $N_{MC} = 10,000$  (corresponding reference values are indicated by black vertical lines); logarithmic absolute difference between PDFs obtained through iES\_ROM and iES\_FSM at points (a) I, (b) II, and (c) III.

Fig. 9 Empirical PDFs of solute concentration at points (a) I, (b) II, and (c) III (see Fig. 1) at the final outer iteration obtained through iES\_ROM considering various values of n (equal to 5 (cyan dashed curve), 10 (magenta), 20 (green), and 30 (blue) for TCs 1, 2, 4, and 6, respectively) and iES\_FSM (red solid curve; results coincide for TCs 1, 2, 4, and 6) when  $N_{MC} = 10,000$  (corresponding reference values are indicated by black lines); logarithmic absolute difference between PDFs obtained through iES\_ROM and iES\_FSM at points (a) I, (b) II, and (c) III.

945

946

Fig. 10 Values of (a)  $E_{\gamma}$ , (b)  $S_{\gamma}$ , and (c)  $E_{obs}$  versus the number of outer iterations obtained through iES\_ROM with n=30 and iES\_FSM considering various values of  $N_{MC}$  (equal to 30, 100, 500, and 10,000 for TCs 13-15 and 6, respectively).

954

Fig. 11 Values of (a)  $E_{x_1}$ , (b)  $E_{x_2}$ , (c)  $E_{q_s}$ , (d)  $S_{x_1}$ , (e)  $S_{x_2}$ , and (f)  $S_{q_s}$  versus the number of outer iterations obtained through iES\_ROM with n=30 and iES\_FSM considering various values of  $N_{MC}$  (equal to 30, 100, 500, and 10,000 for TCs 13-15 and 6, respectively).

Fig. 12 Estimated (ensemble) mean Y fields at the final outer iteration obtained through iES\_ROM with n = 30 (left column) and iES\_FSM (right), considering  $N_{MC} = 30$  (first row), 100 (second), 500 (third), and 10,000 (bottom) for TCs 13-15 and 6, respectively).

Fig. 13 Estimated (ensemble) Y variance fields at the final outer iteration obtained through iES\_ROM with n = 30 (left column) and iES\_FSM (right), considering  $N_{MC} = 30$  (first row), 100 (second), 500 (third), and 10,000 (bottom) (corresponding to TCs 13-15 and 6, respectively).

970971972

Fig. 14 Values of (a)  $E_{\gamma}$ , (b)  $S_{\gamma}$ , and (c)  $E_{obs}$  versus the number of outer iterations obtained through iES\_ROM with n=30 and iES\_FSM considering  $N_m=9$ , 18, and 55 (corresponding to TCs 18, 19, and 6, respectively)

977

Fig. 15 Values of (a)  $E_{x_1}$ , (b)  $E_{x_2}$ , (c)  $E_{q_s}$ , (d)  $S_{x_1}$ , (e)  $S_{x_2}$ , and (f)  $S_{q_s}$  versus the number of outer iterations obtained through iES\_ROM with n=30 and iES\_FSM, considering  $N_m=9$ , 18, and 55 (corresponding to TCs 18, 19, and 6, respectively).

Fig. 16 Empirical PDFs of (a)  $x_{1,q_s}$ , (b)  $x_{2,q_s}$ , and (c)  $\ln q_s$  at the final outer iteration through iES\_ROM with n=30 and iES\_FSM (corresponding reference values indicated by black dashed lines) considering  $N_m=9$ , 18, and 55 (corresponding to TCs 18, 19, and 6, respectively).

Tables

Table 1 Overview of the key settings of the test cases (TCs) analyzed. All TCs are characterized by a zero mean and unit variance of the *Y* reference field;  $\mu$  and  $\sigma_Y^2$  denote the mean and variance of the initial ensemble of the *Y* fields, respectively.

|         | Test Case                         | TC1,                | TC2,                | TC3,      | TC4,      | TC5,    | TC6,    |  |  |  |  |  |
|---------|-----------------------------------|---------------------|---------------------|-----------|-----------|---------|---------|--|--|--|--|--|
|         | Test Case                         | TC7                 | TC8                 | TC9       | TC10      | TC11    | TC12    |  |  |  |  |  |
|         | n                                 | 5                   | 10                  | 15        | 20        | 25      | 30      |  |  |  |  |  |
| Group A | Known                             |                     |                     |           |           |         |         |  |  |  |  |  |
|         | $q_s(\mathbf{x})$ or              | No, Yes             | No, Yes             | No, Yes   | No, Yes   | No, Yes | No, Yes |  |  |  |  |  |
|         | not                               |                     |                     |           |           |         |         |  |  |  |  |  |
|         | Approach                          |                     | iES_FSM and iES_ROM |           |           |         |         |  |  |  |  |  |
|         | Test Case                         | TC13                | TC14                | TC15      | TC6       |         |         |  |  |  |  |  |
| Group B | N <sub>MC</sub>                   | 30                  | 100                 | 500       | 10,000    |         |         |  |  |  |  |  |
|         | Approach                          | iES_FSM and iES_ROM |                     |           |           |         |         |  |  |  |  |  |
|         | Test Case                         | TC16                | TC17                | TC18      | TC19      | TC6     |         |  |  |  |  |  |
| Crown C | $\sigma_{\scriptscriptstyle obs}$ | 0.001               | 0.1                 | 0.01      | 0.01      | 0.01    |         |  |  |  |  |  |
| Group C | $N_{\scriptscriptstyle m}$        | 55                  | 55                  | 9         | 18        | 55      |         |  |  |  |  |  |
|         | Approach                          | iES_FSM and iES_ROM |                     |           |           |         |         |  |  |  |  |  |
|         | Test Case                         | TC20                | TC21                | TC22      | TC23      | TC6     |         |  |  |  |  |  |
|         | μ                                 | -0.5                | 1.5                 | 0.5       | 0.5       | 0.5     |         |  |  |  |  |  |
| Group D | $\sigma_{\scriptscriptstyle Y}^2$ | 1.0                 | 1.0                 | 0.01      | 2.0       | 1.0     |         |  |  |  |  |  |
|         | Approach                          |                     | iE                  | ES_FSM ar | nd iES_RO | M       |         |  |  |  |  |  |
|         |                                   |                     | 58                  |           |           |         |         |  |  |  |  |  |

https://doi.org/10.5194/egusphere-2025-5320 Preprint. Discussion started: 7 November 2025 © Author(s) 2025. CC BY 4.0 License.

|         | Test Case   | TC24    | TC25 | TC26 | TC27 | TC28  | TC6    |  |  |  |
|---------|-------------|---------|------|------|------|-------|--------|--|--|--|
| Group E | $N_{_{sn}}$ | 30      | 100  | 300  | 500  | 1,000 | 10,000 |  |  |  |
|         | Approach    | iES_ROM |      |      |      |       |        |  |  |  |

Table 2 Values of  $E_{\scriptscriptstyle Y}$ ,  $S_{\scriptscriptstyle Y}$ ,  $E_{\scriptscriptstyle obs}$ ,  $E_{\scriptscriptstyle x_1}$ ,  $E_{\scriptscriptstyle x_2}$ ,  $E_{\scriptscriptstyle q_s}$ ,  $S_{\scriptscriptstyle x_1}$ ,  $S_{\scriptscriptstyle x_2}$ , and  $S_{\scriptscriptstyle q_s}$  at the end of

the iteration procedure for TC16, TC6, and TC17 obtained through iES\_ROM and

## 992 iES\_FSM.

|         | TC16      | TC6   | TC17 | TC16      | TC6   | TC17 | TC16      | TC6       | TC17 |  |
|---------|-----------|-------|------|-----------|-------|------|-----------|-----------|------|--|
|         |           | $E_Y$ |      |           | $S_Y$ |      |           | $E_{obs}$ |      |  |
| iES_ROM | 0.41      | 0.41  | 0.53 | 0.50      | 0.50  | 0.62 | 0.01      | 0.02      | 0.07 |  |
| iES_FSM | 0.41      | 0.42  | 0.52 | 0.51      | 0.51  | 0.60 | 0.01      | 0.01      | 0.07 |  |
|         | $E_{x_1}$ |       |      | $E_{x_2}$ |       |      | $E_{q_s}$ |           |      |  |
| iES_ROM | 0.07      | 0.07  | 0.15 | 0.08      | 0.07  | 0.04 | 0.05      | 0.06      | 0.13 |  |
| iES_FSM | 0.02      | 0.02  | 0.06 | 0.13      | 0.12  | 0.09 | 0.05      | 0.05      | 0.11 |  |
|         | $S_{x_1}$ |       |      | $S_{x_2}$ |       |      | $S_{q_s}$ |           |      |  |
| iES_ROM | 0.02      | 0.02  | 0.04 | 0.04      | 0.04  | 0.08 | 0.01      | 0.01      | 0.03 |  |
| iES_FSM | 0.02      | 0.02  | 0.04 | 0.05      | 0.05  | 0.07 | 0.02      | 0.02      | 0.03 |  |

Table 3 Values of  $E_{\scriptscriptstyle Y}$ ,  $S_{\scriptscriptstyle Y}$ ,  $E_{\scriptscriptstyle obs}$ ,  $E_{\scriptscriptstyle x_1}$ ,  $E_{\scriptscriptstyle x_2}$ ,  $E_{\scriptscriptstyle q_s}$ ,  $S_{\scriptscriptstyle x_1}$ ,  $S_{\scriptscriptstyle x_2}$ , and  $S_{\scriptscriptstyle q_s}$  at the end of

the iteration procedure for TC6, TC20, and TC21 obtained through iES\_ROM and

## 996 iES\_FSM.

| Test Case | TC20      | TC6   | TC21 | TC20      | TC6   | TC21 | TC20      | TC6       | TC21 |  |
|-----------|-----------|-------|------|-----------|-------|------|-----------|-----------|------|--|
|           |           | $E_Y$ |      |           | $S_Y$ |      |           | $E_{obs}$ |      |  |
| iES_ROM   | 0.51      | 0.41  | 0.50 | 0.60      | 0.50  | 0.52 | 0.02      | 0.02      | 0.03 |  |
| iES_FSM   | 0.44      | 0.42  | 0.52 | 0.55      | 0.51  | 0.56 | 0.01      | 0.01      | 0.03 |  |
|           | $E_{x_1}$ |       |      | $E_{x_2}$ |       |      | $E_{q_s}$ |           |      |  |
| iES_ROM   | 0.03      | 0.07  | 0.12 | 0.10      | 0.07  | 0.10 | 0.06      | 0.06      | 0.08 |  |
| iES_FSM   | 0.01      | 0.02  | 0.12 | 0.10      | 0.12  | 0.17 | 0.05      | 0.05      | 0.11 |  |
|           | $S_{x_1}$ |       |      | $S_{x_2}$ |       |      | $S_{q_s}$ |           |      |  |
| iES_ROM   | 0.05      | 0.02  | 0.03 | 0.08      | 0.04  | 0.06 | 0.03      | 0.01      | 0.01 |  |
| iES_FSM   | 0.03      | 0.02  | 0.04 | 0.06      | 0.05  | 0.06 | 0.02      | 0.02      | 0.02 |  |

Table 4 Values of  $E_{\scriptscriptstyle Y}$ ,  $S_{\scriptscriptstyle Y}$ ,  $E_{\scriptscriptstyle obs}$ ,  $E_{\scriptscriptstyle x_1}$ ,  $E_{\scriptscriptstyle x_2}$ ,  $E_{\scriptscriptstyle q_s}$ ,  $S_{\scriptscriptstyle x_1}$ ,  $S_{\scriptscriptstyle x_2}$ , and  $S_{\scriptscriptstyle q_s}$  at the end of

the iteration procedure for TC6, TC22, and TC23 obtained through iES\_ROM and

## 1000 iES\_FSM.

| Test Case | TC22           | TC6   | TC23 | TC22      | TC6   | TC23 | TC22      | TC6       | TC23 |  |
|-----------|----------------|-------|------|-----------|-------|------|-----------|-----------|------|--|
|           |                | $E_Y$ |      |           | $S_Y$ |      |           | $E_{obs}$ |      |  |
| iES_ROM   | 0.47           | 0.41  | 0.46 | 0.06      | 0.50  | 0.77 | 0.01      | 0.02      | 0.02 |  |
| iES_FSM   | 0.43           | 0.42  | 0.47 | 0.06      | 0.51  | 0.80 | 0.01      | 0.01      | 0.01 |  |
|           | $E_{x_{ m l}}$ |       |      | $E_{x_2}$ |       |      | $E_{q_s}$ |           |      |  |
| iES_ROM   | 0.07           | 0.07  | 0.09 | 0.15      | 0.07  | 0.12 | 0.02      | 0.06      | 0.08 |  |
| iES_FSM   | 0.02           | 0.02  | 0.05 | 0.22      | 0.12  | 0.17 | 0.02      | 0.05      | 0.08 |  |
|           | $S_{x_1}$      |       |      | $S_{x_2}$ |       |      | $S_{q_s}$ |           |      |  |
| iES_ROM   | 0.003          | 0.02  | 0.05 | 0.004     | 0.04  | 0.09 | 0.003     | 0.01      | 0.02 |  |
| iES_FSM   | 0.002          | 0.02  | 0.04 | 0.003     | 0.05  | 0.08 | 0.003     | 0.02      | 0.03 |  |

Table 5 Percentage differences between the values of the selected metrics (i.e.,  $E_{\rm y}$ ,

$S_{Y}$ ,  $E_{obs}$ ,  $E_{x_1}$ ,  $E_{x_2}$ ,  $E_{q_s}$ ,  $S_{x_1}$ ,  $S_{x_2}$ , and  $S_{q_s}$ ) at the end of the iteration procedure

for TCs 24-28 obtained through iES\_ROM (values corresponding to TC6 are taken as

references).

| Test Case | $E_Y$ | $S_Y$ | $E_{obs}$ | $E_{_{X_{_{\mathrm{l}}}}}$ | $S_{x_1}$ | $E_{x_2}$ | $S_{x_2}$ | $E_{q_s}$ | $S_{q_s}$ |
|-----------|-------|-------|-----------|----------------------------|-----------|-----------|-----------|-----------|-----------|
| TC24      | 11.88 | 6.34  | 21.60     | 44.17                      | 58.50     | 25.04     | 34.75     | 32.29     | 55.20     |
| TC25      | 10.16 | 3.66  | 6.76      | 27.83                      | 35.54     | 13.24     | 9.58      | 25.01     | 37.58     |
| TC26      | 7.40  | 1.97  | 13.00     | 22.10                      | 16.89     | 35.54     | 11.26     | 13.06     | 15.09     |
| TC27      | 4.58  | 0.14  | 2.66      | 11.42                      | 0.17      | 22.93     | 1.22      | 17.74     | 3.37      |
| TC28      | 0.50  | 0.21  | 4.18      | 17.19                      | 1.33      | 31.33     | 5.86      | 14.03     | 0.07      |

1007