# Peer review of "Joint characterization of heterogeneous conductivity fields and pumping well attributes through iterative ensemble smoother with a reduced-order modeling strategy for solute transport 3 4 Chuan-An Xia1, Jiayun Li2\*, Bill X. Hu3, Alberto Guadagnini4,5,"

_EGUsphere, 2025_

## Author Comment (AC1)

Referee#1

The paper presents a solid and carefully executed study on coupling a POD-based reduced order transport model with an iterative ensemble smoother for joint estimation of K and well attributes. Results are convincing, and the work is publishable after some focused improvements.

Answer: We are grateful for the positive and constructive comments. We have carefully revised the manuscript and provided our responses to each of the comments emerged. The line numbers provided correspond to the clean (without tracked changes) version of the revised manuscript.

Improve the novelty statement in the Introduction and Conclusions; what is new relative to existing ROM+DA studies: (i) joint estimation of heterogeneous K and hidden pumping well attributes, (ii) reduction of only the transport equation while keeping flow full-order, and (iii) the systematic multi-factor analysis (ROM size, ensemble size, prior stats, noise, snapshot size).

Answer: It is worth to note that only a few studies are devoted to performing uncertainty quantification relying on a reduced order model (ROM) approach for solute transport. To support this statement, we write: (lines: 171-176) "Although conceptual insights can be drawn from ROMC studies addressing groundwater flow (e.g., Pasetto et al., 2014; Xia et al., 2020, 2025), influence of key factors (such as, e.g., dimensionality of the reduced concentration space and strength of hydraulic conductivity heterogeneity) on accuracy and robustness of ROMC-based UQ still remains poorly characterized". We then state that: (lines: 177-179) "Building upon these works, the present study introduces a novel framework that integrates the iES with a ROM for solute transport (hereafter referred to as iES_ROM)".

We further emphasize that: (lines: 60-62) "Despite the relevance of these issues, only limited research has been devoted to the identification and quantification of pumping rates and spatial locations of such hidden wells.".

Additionally, we point out that the scenarios we consider include the key factors that can have an impact on ROM performance in these contexts and are designed to examine the potential of the novel framework we propose.

Provide concrete numbers and protocol: how many realizations and time levels are used, whether snapshots come from prior draws or a single reference field, and whether their statistics match those used in the DA experiments. Add a short discussion of how ROM performance might change if the prior used for snapshot generation differs from that used in assimilation. Add also a short discussion (no need for new runs) on how sensitive the ROM is if the prior used for snapshot generation differs from the prior used for DA.

Answer: We have listed the detailed settings for each test case in Table 1. We now write that: (lines: 353-355) "A uniform time step of 1 day is considered, our analyses encompassing a total simulation time of 10 days (i.e., $T_s = 10$ days and $N_t = 10$).".

Snapshots are taken as FSM solutions corresponding to $m$ randomly selected realizations of the initial ensemble of long-conductivity ($Y$) fields. Our revised manuscript then states that: (lines: 267-273) "The basis functions forming the entries of $\mathbf{P}$ are computed as the leading eigenvectors (corresponding to the highest eigenvalues) of the covariance of solute concentration evaluated through $N_{sn}$ numerical solutions (i.e., $\mathbf{c}^1$, $\mathbf{c}^2$, ..., and $\mathbf{c}^{N_{sn}}$) of the FSM. Here, $N_{sn} = m \times N_t$, where $m$ is the number of MC realizations of hydraulic conductivity that are randomly sampled from the initial ensemble of $Y$ fields, each yielding $N_t = T_s / \Delta t$ ($\Delta t$ corresponding to a uniform time step) numerical solutions of Equation (2)."

Concerning the discussion for the influence of snapshots on the accuracy of ROM solution, we now write that: (lines: 686-694) "Additionally, we emphasize that relying on realizations of $Y$ associated with (spatial) statistics different from their theoretical counterparts linked to the initial ensemble of $Y$ fields can contribute to deteriorate the quality of the selected snapshots. Low quality snapshots yield low quality basis functions and low accuracy of ROM outcomes (see our results in Section 4.1; Pasetto et al., 2014; Xia et al., 2020). The latter deteriorate the accuracy of conductivity estimates and pumping well attributes. Additional studies should be devoted to assess the potential of techniques (including, e.g., greedy algorithms) that might contribute to increase the quality of snapshots.".

State whether snapshots are mean-centered, and discuss briefly how omitting a separate mean field affects accuracy. Add a short justification of why a "mean + anomalies" representation is less convenient in your iES implementation, and whether it might reduce ROM error.

Answer: As we state in the original manuscript (lines: 300-306), "The degree of compatibility of ROM to iES is reduced when considering a typical Karhunen-Loève expansion of $\mathbf{c}^i$ (i.e., $\mathbf{c}^i \approx \langle \mathbf{c} \rangle + \sum_{j=1}^{n} \alpha_j^i \mathbf{p}_j = \langle \mathbf{c} \rangle + \mathbf{P}\boldsymbol{\alpha}^i$ ). This is related to the observation that $\langle \mathbf{c} \rangle$ evolves with time and needs to be evaluated at each time step. This, in turn, implies that $m$ numerical solutions of solute concentration through FSM need to be obtained to evaluate $\langle \mathbf{c} \rangle$ at every outer iteration of iES. Hence, computational advantages of employing ROM are reduced while coding complexity increases.".

Make clear which ensemble sizes are realistic for applied hydrogeological problems (e.g. a few hundred to 1000), and present N_MC = 10000 explicitly as a reference benchmark. Emphasize results and cost accuracy trade offs for the practically relevant range.

Answer: Quantification of feasible and acceptable ensemble sizes is case-dependent. When considering a field scenario, one would need to find a trade-off that corresponds to relatively low computational costs while keeping acceptable accuracy of parameter estimates. We now write in our Conclusions that: (lines: 761-764) "Moreover, the values of $N_{MC}$ that one should consider in a field application are case-dependent. In this context, localization techniques can be embedded in DA processes, as these can reduce negative influences of spurious correlation on parameter estimate arising from reliance on small ensemble sizes.".

Use Tables 1–5 and possibly a small schematic/flowchart to clearly show what each group (A–E) varies and why. In the results, slightly condense repetitive descriptions and highlight cross-group patterns and any non-intuitive behaviors (e.g. non-monotonic trends).

Answer: We recall that we have clearly introduced the settings for each group in Section 3. The latter reads: (lines: 384-420) "To explore the potential of iES_ROM, several showcases are designed to highlight key features of interest. Five groups of test cases (TCs) are designed and organized as detailed in the following (see also Table 1).

➤ **Group A**. It includes twelve TCs (i.e., TC1-TC12), enabling us to compare performances of iES_FSM and iES_ROM associated with diverse values of $n$ when the pumping rate and locations are either known (TC1-TC6) or unknown (TC7-TC12). The dimension of the ROM is considered equal to {5, 10, 15, 20, 25, 30}, these values being consistent with those most commonly analyzed in previous studies (Pasetto et al., 2014; Xia et al., 2020, 2025).

➤ **Group B**. It includes four TCs (i.e., TC6 and TC13-TC15), enabling us to compare the performances of iES_FSM and iES_ROM with the largest value of $n$ analyzed (i.e, $n = 30$) and considering diverse values of $N_{MC}$ corresponding to {30, 100, 500, 10,000}. The latter are values of $N_{MC}$ commonly tested in previous studies (Chen and Zhang, 2006; Xia et al., 2021, 2024).

➢ **Group C**. It includes five TCs (i.e., TC6 and TC16-TC19), designed to analyze the ability of iES_ROM to cope with diverse quality and quantity of available measurements. Performances of iES_FSM and iES_ROM are also compared when $\sigma_{obs}$ = {0.001, 0.01, 0.1} and the number of observation locations corresponds to a value selected from {9 (Fig. 1b), 18 (Fig. 1c), 55 (Fig. 1d)}.

➢ **Group D**. It includes five TCs (i.e., TC6 and TC_20-TC23), enabling us to study the effect of $\mu$ and $\sigma_Y^2$ of the initial ensemble of $Y$ on the accuracies of estimates of conductivity and pumping rate and well location through iES_FSM and iES_ROM. Values of $\mu$ and $\sigma_Y^2$ of the initial ensemble of $Y$ fields are selected from {-0.5, 1.2, 2.0} and {0.01, 1.0, 2.0}, respectively.

➢ **Group E**. It includes six TCs (i.e., TC6 and TC24-TC28), with the aim of investigating the effect of $N_{sn}$ on the accuracies of the estimation of conductivity and well pumping rate and location through iES_ROM and on computation time requirements. Values of $N_{sn}$ in TC24-TC28 and TC6 are equal to 30, 100, 300, 500, 1,000, and 10,000, respectively.

Note that, without specified otherwise, default settings for the above mentioned TCs correspond to TC6 which is designed with $n$ = 30, $N_{MC}$ = 10,000, $N_{sn}$ = 10,000, $N_m$ = 55, $\sigma_{obs}$ = 0.01, and values of $\mu$ and $\sigma_Y^2$ of the initial ensemble of $Y$ equal to 1.2 and 1.0, respectively. Except for TC8-TC12, the source/sink term is associated with uncertainty.".

Prompted by the reviewer's comment, we also slightly condense repetitive descriptions and highlight cross-group patterns and any non-intuitive behaviors in our revised manuscript.

In the Conclusions, clearly delimit the domain of validity: 2D confined aquifer, steady-state flow, single non-reactive solute, single well. Briefly comment on expected challenges and required modifications for transient flow, multiple wells, or reactive/density-dependent transport.

Answer: We state (at the beginning of our Conclusions) that: "This study addresses joint estimation of (uncertain, spatially heterogeneous) hydraulic conductivities and attributes (location and flow rate) of a pumping well in a two-dimensional confined aquifer in the presence of (non-reactive) solute transport taking place across a steady-state flow field.".

We now add at the end of our Conclusions: "Additional elements of interest associated with future studies on coupling iES with ROM include the analysis of transient saturated/unsaturated flow, reactive transport, and density-dependent flow/transport scenarios. When considering nonlinear systems, reliance on discrete matrix interpolation schemes (Negri et al., 2015; Bonomi et al., 2017) constitutes a promising approach to enhance computational advantages of ROM.

**References**

Bonomi, D., Manzoni, A., Quarteroni, A., 2017. A matrix DEIM technique for model reduction of nonlinear parametrized problems in cardiac mechanics. Comput. Methods Appl. Mech. Eng. 324, 300-326.

Negri, F., Manzoni, A., Amsallem, D., 2015. Efficient model reduction of parametrized systems by matrix discrete empirical interpolation. J. Comput. Phys. 303, 431-454.

Overall, these changes are mostly clarifications and presentation refinements; the core methodology and results appear sound.

Answer: We thank the reviewer for the very positive appraisal of our work.

---

## Author Comment (AC2)

Referee#2

Based on the **HESS principal criteria** (scale **1 = Excellent, 2 = Good, 3 = Fair, 4 = Poor**).

Answer: We thank the reviewer for the positive and constructive feedback. We have carefully considered all comments, including those submitted through the review system, and have addressed each point in our responses below. With reference to the parts denoted, e.g., as "Presentation quality: **2 (Good)**", our responses are designed to address the overall elements emerged from the comments of the reviewer instead of providing a response to each individual comment which sometimes is an evaluation from the reviewer on a specified aspect of the manuscript. The line numbers provided correspond to the clean (without tracked changes) version of the revised manuscript.

Scientific significance: **2 (Good)**

The manuscript provides a substantive contribution to ROM–DA coupling for groundwater systems by proposing iES_ROM for the **joint estimation** of **hidden pumping-well attributes** (rate and location), and by demonstrating it through a broad **multi-factor sensitivity assessment** (28 test cases). The contribution is clearly relevant to the scope of HESS; however, the manuscript should state more explicitly—both in the Introduction and Conclusions—what is novel relative to existing ROM+DA studies to firmly support an "Excellent" rating.

Answer: We state that: (lines: 171-176) "Although conceptual insights can be drawn from ROMC studies addressing groundwater flow (e.g., Pasetto et al., 2014; Xia et al., 2020, 2025), influence of key factors (such as, e.g., dimensionality of the reduced concentration space and strength of hydraulic conductivity heterogeneity) on accuracy and robustness of ROMC-based UQ still remains poorly characterized" and that: (lines: 177-179) "Building upon these works, the present study introduces a novel framework that integrates the iES with a ROM for solute transport (hereafter

referred to as iES_ROM). The ensuing framework enables one to efficiently quantify uncertainty and jointly estimate system parameters in groundwater-related modeling scenarios.".

Scientific quality: **2 (Good)**

The overall scientific approach is sound: iES is clearly formulated, the POD/Galerkin transport ROM is properly derived, the method is benchmarked against **iES_FSM**, and the discussion acknowledges nonlinearity and occasional non-monotonic behavior (especially for well attributes). The literature coverage is generally appropriate, and the discussion is balanced. That said, **traceability and reproducibility** would be strengthened by a more concrete snapshot-generation protocol (how snapshots are selected, whether they are mean-centered, and a brief discussion of sensitivity to "prior mismatch"), which would further reinforce methodological robustness.

Answer: We thanks for the reviewer for the positive comments.

Our revised manuscript states that: (lines: 267-273) "The basis functions forming the entries of **P** are computed as the leading eigenvectors (corresponding to the highest eigenvalues) of the covariance of solute concentration evaluated through $N_{sn}$ numerical solutions (i.e., $\mathbf{c}^1$, $\mathbf{c}^2$, ..., and $\mathbf{c}^{N_{sn}}$) of the FSM. Here, $N_{sn} = m \times N_t$, where $m$ is the number of MC realizations of hydraulic conductivity that are randomly sampled from the initial ensemble of $Y$ fields, each yielding $N_t = T_s / \Delta t$ ($\Delta t$ corresponding to a uniform time step) numerical solutions of Equation (2)."

As we state in the original manuscript (lines: 300-306), "The degree of compatibility of ROM to iES is reduced when considering a typical Karhunen-Loève expansion of $\mathbf{c}^i$ (i.e., $\mathbf{c}^i \approx \langle \mathbf{c} \rangle + \sum_{j=1}^{n} \alpha_j^i \mathbf{p}_j = \langle \mathbf{c} \rangle + \mathbf{P}\boldsymbol{\alpha}^i$). This is related to the observation that $\langle \mathbf{c} \rangle$ evolves with time and needs to be evaluated at each time step. This, in turn, implies that $m$ numerical solutions of solute concentration through FSM

need to be obtained to evaluate $\langle \mathbf{c} \rangle$ at every outer iteration of iES. Hence, computational advantages of employing ROM are reduced while coding complexity increases.".

Concerning the discussion for the influence of snapshots on the accuracy of ROM solution, we now write that: (lines: 686-694) "Additionally, we emphasize that relying on realizations of $Y$ associated with (spatial) statistics different from their theoretical counterparts linked to the initial ensemble of $Y$ fields can contribute to deteriorate the quality of the selected snapshots. Low quality snapshots yield low quality basis functions and low accuracy of ROM outcomes (see our results in Section 4.1; Pasetto et al., 2014; Xia et al., 2020). These elements, in turn, contribute to deteriorate the accuracy of conductivity estimates and pumping well attributes. Additional studies should be devoted to assess the potential of techniques (including, e.g., greedy algorithms) that might contribute to increase the quality of snapshots.".

Presentation quality: **2 (Good)**
The presentation is clear and well-structured; the tables (e.g., Table 1, which serves as a roadmap for Groups A–E) and figures adequately support the conclusions. The English is generally correct and technically appropriate. Suggested improvements include condensing repetitive result descriptions, emphasizing cross-group patterns and non-intuitive behaviors (e.g., non-monotonic trends), and adding a small workflow schematic/flowchart for iES_ROM (since the POD basis is built once and reused).

Answer: We now condense repetitive descriptions and explanations. Table 1 lists details of the five Groups analyzed, each corresponding to the analysis of a specified factor with the aim of testing the potential of iES_ROM. We have emphasized the non-monotonic trends of key metrics versus the number of outer iterations. We now add the following flowchart to clarify the conceptual and operational framework underpinning iES_ROM. We do hope these elements contribute to improve clarity.

[Figure]

Fig. 1 Workflow of iES_ROM, comprising (*i*) standard MC simulation of groundwater flow (relying on FSM), (*ii*) reduced-order MC approach for solute transport (relying on ROM), and (*iii*) iES coupled with ROM.

1. **Does the paper address relevant scientific questions within the scope of HESS?**

   Yes. The manuscript tackles a core HESS-relevant problem: data assimilation and uncertainty quantification in groundwater systems, combining steady-state flow and transient solute transport to infer hydrologic/hydrogeologic properties and unobserved stresses. Specifically, it targets the joint identification of a heterogeneous conductivity field $Y = \ln K$ and hidden pumping-well attributes (rate and location) from head and concentration observations.

   Answer: We thank the reviewer for the positive assessment.

2. **Does the paper present novel concepts, ideas, tools, or data?**

   Yes, although the novelty should be stated more explicitly in the Introduction and Conclusions. The main novelties are: (i) joint estimation of heterogeneous Y=lnK and hidden well attributes (rate $q_s$ and coordinates), (ii) a hybrid strategy that reduces only the transport equation (POD-ROM) while keeping steady-state flow full order, and (iii) a systematic multi-factor evaluation with 28 test cases (Groups A–E) spanning ROM dimension, ensemble size, observation noise and network density, prior statistics, and snapshot size.

   Answer: We thank the reviewer for the positive assessment.

3. **Are substantial conclusions reached?**

   Yes. The paper concludes that iES_ROM can achieve accuracy close to the full-system benchmark iES_FSM within the tested setting, while substantially reducing CPU cost. It supports this with multi-group results and a concrete CPU-time comparison (e.g., the reported TC6 vs TC28 timing).

   Answer: We thank the reviewer for the positive assessment.

4. **Are the scientific methods and assumptions valid and clearly outlined?**

   Mostly yes. The iES formulation, the joint parameter vector (spatial field plus well parameters), and the POD/Galerkin ROM derivation for transport are clearly presented, and the comparison against iES_FSM is a strong methodological choice. The physical assumptions (2D confined aquifer, steady-state flow, non-reactive transport, observation noise) are stated. Remaining points that would benefit from clarification are whether snapshots are mean-centered and a more explicit discussion of the implications of not using a "mean + anomalies" representation.

   Answer: We thank the reviewer for the positive assessment. We refer to our answers above related to this point.

5. **Are the results sufficient to support the interpretations and conclusions?**

   Yes. The manuscript provides structured evidence across Groups A–E, with summary tables, convergence/accuracy metrics, and distributional comparisons (PDFs and KLDs) that support the claim that ROM can closely match FSM when the ROM dimension and snapshot size are adequate. The discussion also acknowledges non-monotonic behaviors, which strengthens interpretive balance.

   Answer: We thank the reviewer for the positive assessment.

6. **Is the description of experiments and calculations sufficiently complete and precise to allow reproduction (traceability of results)?**

   Partly. Many key ingredients are specified (domain and boundary conditions, mesh size, time step, simulation horizon, test-case matrix, and the definition $N_{sn}$, $m$, and $N_t$). However, full traceability would be improved by an explicit operational snapshot protocol: how many realizations are selected in practice (the text indicates "arbitrarily chosen"), whether snapshots come from prior draws or a reference field, whether snapshot-generation statistics match those used in DA, and a brief discussion of expected sensitivity to prior mismatch (snapshot prior vs assimilation prior).

   Answer: We thank the reviewer for the positive assessment. We refer to our answers above addressing the reviewer's comment.

7. **Do the authors give proper credit to related work and clearly indicate their own new/original contribution?**

   Generally, yes in terms of literature coverage and positioning. The manuscript cites relevant ROM/ROMC and DA foundations and explains how iES_ROM is constructed and benchmarked. Still, the paper would benefit from a concise "what is new compared to existing ROM+DA studies" paragraph with a few closely related citations, so the original contribution is unmistakable.

8. **Does the title clearly reflect the contents of the paper?**

Yes. The title accurately reflects joint characterization, IES-based estimation, and a reduced-order strategy specifically for solute transport, as well as the inclusion of pumping-well attributes.

Answer: We thank the reviewer for the positive assessment.

9. **Does the abstract provide a concise and complete summary?**

Yes. It describes the problem, the proposed iES_ROM method, and the iES_FSM benchmark, the scope of the test campaign, and the key takeaways on recommended ROM dimensions/ensemble sizes and computational savings. A minor improvement would be to state the validity domain (2D confined, steady-state flow, single non-reactive solute, single well) explicitly in one sentence.

Answer: We thank the reviewer for the positive assessment. We refer to our answers above addressing the reviewer's comment.

10. **Is the overall presentation well structured and clear?**

Yes. The manuscript follows a logical flow from methods to test design to results, organized by Groups A–E, and is supported by a functional "roadmap" table. The presentation could be further improved by more explicit signposting to Table 1 at the start of each group subsection, by condensing repetitive descriptions, and by highlighting cross-group patterns and non-intuitive behaviors.

Answer: We thank the reviewer for the positive assessment. We refer to our answers above addressing the reviewer's comment.

11. **Is the language fluent and precise?**

    Yes overall. The English is technically appropriate and generally clear. Minor consistency edits (terminology and symbols) would further improve precision.

    Answer: We thank the reviewer for the positive assessment. A detailed consistency check has been performed.

12. **Are mathematical formulae, symbols, abbreviations, and units correctly defined and used?**

    Mostly yes. Key quantities ($Y=\ln K$, $N_{MC}$, $\sigma_{obs}$, $N_m$, $N_{sn}$) and the ROM/iES formulations are defined and used consistently in the core derivations, and the manuscript notes consistent units. Minor issues include standardizing notation (e.g., $N_{MC}$ vs $N_{mc}$) and clarifying whether "time levels" include the initial condition or only simulated steps, which affects $N_t$ and thus $N_{sn}$.

    Answer: We thank the reviewer for the positive assessment. Consistency of notation has been checked in details.

13. **Should any parts of the paper be clarified, reduced, combined, or eliminated?**

    Yes. The main clarifications are the snapshot protocol, mean-centering, and a short discussion of prior mismatch. In Results, some case-by-case narratives can be reduced/combined by summarizing each group with a compact set of key findings and then highlighting the few non-intuitive outcomes (e.g., small-ensemble inbreeding and non-monotonic trends for well attributes). Adding a small workflow schematic/flowchart of iES_ROM would also improve clarity without inflating text length.

    Answer: We thank the reviewer for the positive assessment. We refer to our answers above addressing the reviewer's comment.

14. **Are the number and quality of references appropriate?**

Yes. The manuscript cites relevant ROM/POD/ROMC and DA literature and provides an adequate methodological context. Strengthening the novelty statement with a handful of the closest ROM+DA references would further improve the framing.

Answer: We thank the reviewer for the positive assessment. We refer to our answers above addressing the reviewer's comment.

15. **Is the amount and quality of supplementary material appropriate?**

Yes. The supplement is used appropriately to support distributional comparisons (e.g., KLD-based agreement between iES_ROM and iES_FSM) and to provide additional supporting results that would otherwise distract from the main narrative. A slight improvement would be a brief sentence in the main text indicating what each supplementary table contributes and when the reader should consult it.

Answer: We thank the reviewer for the positive assessment. We refer to our answers above addressing the reviewer's comment.

**Technical and typographical suggestions (minor)**

1) Ensure consistent notation for ensemble size (e.g., $N_{MC}$ vs $N_{mc}$)

Answer: We have carefully checked the whole text and made sure that $N_{MC}$ is the symbol that is consistently used to denote ensemble size.

2) Clearly define whether the number of time levels $N_t$ includes the initial condition or only the simulated time steps.

Answer: We now write that: (lines: 353-355) "A uniform time step of 1 day is considered for a total simulation time of 10 days (i.e., $T_s = 10$ days and $N_t = 10$).".

3) Where CPU time is used to support the efficiency claim, consider presenting it once in the Results in a compact way (e.g., a short table or a concise paragraph) and referencing it in the Conclusions.

   Answer: We now write that: (lines: 677-682) "A CPU time of about 13 minutes is required for running TC28 (using a processor 13th Gen Intel(R) Core(TM) i7-13700K 3.40 GHz with 32 GB RAM). The CPU time required to complete TC6 upon relying on iES_FSM (122 minutes) is about 9 times the corresponding CPU time required to complete TC28 through iES_ROM (28 minutes), percentage differences associated with $E_Y$ and $S_Y$ being equal to 0.50% and 0.21%, respectively.".

4) Double-check symbol definitions at first use (e.g., $n$, $N_{sn}$, $N_m$, $\sigma_{obs}$) and keep units explicit where relevant.

   Answer: We have carefully checked the whole manuscript to this end. We further state that (lines: 344-345) "Here and hereafter, all quantities are given in consistent (length/mass/time) units.".

5) If distributional comparisons (e.g., KLD in supplementary tables) are key to the argument, add one sentence in the main text explaining what the supplement contributes.

   Answer: We have written that (lines: 527-542) "As a complement to these results, values of the Kullback-Leibler Divergence (KLD) between the (sample) PDFs of head at the three reference points at the last outer iteration obtained through iES_FSM ($h_{FSM}$) and iES_ROM ($h_{ROM}$) with $n = 5$ (TC1), 10 (TC2), 20 (TC4), and 30 (TC6) are listed in Table S1 (see supplementary information). We recall that values of KLD($h_{ROM}\|h_{FSM}$) (or (KLD($h_{FSM}\|h_{ROM}$)) quantify (in a global sense) information loss when using $h_{FSM}$ ($h_{ROM}$) to approximate $h_{ROM}$ ($h_{FSM}$). Values of KLD($h_{ROM}\|h_{FSM}$) generally increase with $n$. This indicates that the difference between PDFs of $h_{ROM}$ and $h_{FSM}$ decrease as $n$ increases. While the highest values of KLD($h_{FSM}\|h_{ROM}$) correspond to $n = 5$, no clear decreasing trends with increasing $n$ are observed. Furthermore, the difference between KLD($h_{ROM}\|h_{FSM}$) and KLD($h_{FSM}\|h_{ROM}$)

generally decreases as $n$ increases. This is related to the observation that the accuracy of ROM tends to increase as the dimension of the reduced-order model increase. Values of KLD between the empirical PDFs of solute concentrations at the three selected reference points at the last outer iteration obtained through iES_FSM ($c_{FSM}$) and iES_ROM ($c_{ROM}$) with $n = 5$ (TC1), 10 (TC2), 20 (TC4), and 30 (TC6) are listed in Table S2 (see supplementary information).".